# Using Noise to Infer Aspects of Simplicity Without Learning

**Zachery Boner**[1]* **Harry Chen**[1]* **Lesia Semenova**[2]* **Ronald Parr**[1] **Cynthia Rudin**[1]

[1]Department of Computer Science, Duke University, [2]Microsoft Research

zachery.boner@duke.edu, harry.chen084@duke.edu, lsemenova@microsoft.com,
ronald.parr@duke.edu, cynthia.rudin@duke.edu

## Abstract

Noise in data significantly influences decision-making in the data science process. In fact, it has been shown that noise in data generation processes leads practitioners to find simpler models. However, an open question still remains: what is the degree of model simplification we can expect under different noise levels? In this work, we address this question by investigating the relationship between the amount of noise and model simplicity across various hypothesis spaces, focusing on decision trees and linear models. We formally show that noise acts as an implicit regularizer for several different noise models. Furthermore, we prove that Rashomon sets (sets of near-optimal models) constructed with noisy data tend to contain simpler models than corresponding Rashomon sets with non-noisy data. Additionally, we show that noise expands the set of "good" features and consequently enlarges the set of models that use at least one good feature. Our work offers theoretical guarantees and practical insights for practitioners and policymakers on whether simple-yet-accurate machine learning models are likely to exist, based on knowledge of noise levels in the data generation process.

## 1 Introduction

Machine learning (ML) is being used more and more for high-stakes decisions, and there is a need for clear policy guidance. Simple models have advantages: they are much easier to troubleshoot and use. However, there is a concern that they are not as accurate as more complex black box models, which makes it challenging to provide guidance for policy makers to recommend simple models.

We believe we have barely scratched the surface of understanding simplicity in machine learning. Recent work suggests we consider the *Rashomon Effect* [Breiman, 2001], which is the phenomenon that datasets often admit many different good models. Semenova et al. [2022] shows that when there are a lot of good models, some of them are probably simple, meaning that there is no accuracy-simplicity trade-off. But we do not know in advance *how* simple these models can get. For some applications, very sparse additive models or decision trees perform as well as the best black box models; these datasets do not benefit from complex models at all. When does that happen? How extreme on the simplicity scale do we expect these models to go?

A key insight into this question comes from Semenova et al. [2023], who showed that the simplicity we can expect seems to be related to the amount of outcome noise in the data generation process, which we denote informally as "$\rho$." When $\rho$ is larger, we can get simpler models with performance comparable to the best models. However, while Semenova et al. [2023] showed that noise levels are important, they did not provide a quantitative relationship between noise and simplicity.

In this work, we ask a question that allows us to better understand the connections among noise in the data generation processes, model simplicity, and the Rashomon Effect. The question is as

---

*These authors contributed equally to this work. The names are listed in alphabetical order.

38th Conference on Neural Information Processing Systems (NeurIPS 2024).

follows: Given the noise level $\rho$, how much simpler can our ML models get as compared to the non-noisy case while still maintaining similar generalization performance? Semenova et al. [2023] proves only that the hypothesis space could be simplified in the presence of noise, but does not discuss how much.

To answer this question, we formally prove for two common hypothesis spaces that *noise is an implicit regularizer*, thus leading to simpler models. We quantify how much regularization is added to models as a precise function of the amount of noise, $\rho$, for several types of noise and regularization. Table 1 summarizes the main results of this paper. For various combinations of hypothesis spaces, losses, and types of noise, we show that if we have regularization $\lambda$ and noise $\rho$, the optimization problem on noisy data is equivalent to optimizing over the cleaner data with stronger regularization (described in the rightmost column). This means that if we have noisy data, we get a simpler model than if the data were cleaner and we had performed the same optimization.

Since the Rashomon Effect seems to be an important mechanism to understand simplicity, we also study how noise affects it. The *Rashomon set* is the set of models that have comparable performance to the best models within a class. In this work, we show that, in the presence of noise, the Rashomon set tends to consist of simpler models than in the non-noisy setting. This means that if a user is looking for a model in the Rashomon set that obeys specific constraints (e.g., fairness or monotonicity), these models will be simpler (and this task is likely to be easier) in the presence of noise.

We also study how noise changes the relationship between features and outcomes in an unregularized setting. Specifically, we show that, for decision trees, the number of "good features" (having high AUC relative to other features) increases with noise, and the set of models that use at least one good feature grows larger. Since most models in the Rashomon set use at least one good feature, the Rashomon set of the unregularized hypothesis space might also increase in size with increased noise. This attempts to shed more light on the results that previous work [Semenova et al., 2023] only observed empirically.

We confirm our results empirically and provide practical guidance for the datasets from domains of criminal justice and lending, where we expect outcome noise due to the random nature of the data generation process. We hope that our results are the initial steps that will help machine learning practitioners, and possibly policymakers, to reason about the simplicity of models they can expect to encounter for many high-stakes decision domains.

Table 1: Summary of paper contributions and answers to the key question.

| | Complex hypothesis space/model | Loss | Noise, $\rho$ | Effective regularization |
|---|---|---|---|---|
| 1 | Any model optimized on regularized 0-1 loss (e.g. sparse decision trees with leaf penalty, rule lists with length penalty, scoring systems with sparsity penalty) with regularization penalty $\lambda$ | Misclassification error | Random label noise | Model optimized with regularization penalty $\frac{\lambda}{1-2\rho}$ |
| 2 | Linear models | Exponential loss | Additive attribute noise | Linear models that minimize logarithm of exponential loss with $\ell_2$ regularization, where $\frac{1}{2}\rho^2$ is the regularization parameter |

## 2 Related Work

There are several bodies of related literature.

**Rashomon sets.** The Rashomon set – the set of all near-optimal models – has been studied primarily in the context of its usefulness for solving downstream problems. Examples include developing stable measures of variable importance [Donnelly et al., 2023, Dong and Rudin, 2020, Fisher et al., 2019, Smith et al., 2020], quantifying predictive multiplicity [Marx et al., 2020, Hsu and Calmon, 2022, Watson-Daniels et al., 2023], and understanding fairness [Aïvodji et al., 2021, Coston et al., 2021, Shamsabadi et al., 2022]. There are also algorithms for computing complete or approximate Rashomon sets for a variety of hypothesis spaces [Mata et al., 2022, Xin et al., 2022, Zhong et al., 2023]. The most related prior work to this paper is the work of Semenova et al. [2022] and Semenova et al. [2023], which together demonstrate the existence of large Rashomon sets, and therefore simple models, when there is a significant amount of randomness in the data generation process. In

comparison to prior work, ours is the first to provide a quantitative relationship between noise levels and quantities related to simplicity such as regularization and the contents of the Rashomon set.

**Policy and interpretable ML.** With new regulations including "right to explanation," users can request an explanation if an automated decision has been made about them. However, such explanations are often post-hoc and may be misleading [Rudin et al., 2022, Rudin, 2019, Han et al., 2022, Adebayo et al., 2018], contradictory [Krishna et al., 2022], incomplete [Rudin, 2019], or failing in adversarial contexts [Bordt et al., 2022]. Interpretable models do not have these problems, and empirically, interpretable models in high-stakes decision domains tend to be as accurate as black-box models; this has been shown in lending [e.g., Chen et al., 2022], criminal justice [e.g., Angelino et al., 2017], and healthcare [e.g., Zhu et al., 2023]. However, policy makers still permit black boxes for high-stakes domains, possibly based on accuracy-simplicity trade-off concerns. Thus, more evidence about when this trade-off does and does not exist will be helpful.

**Noise and regularization.** The influence of noise on regularization has been studied for the hypothesis space of neural networks, though no prior work is directly relevant to our aims. Bishop [1995] showed that injecting a small amount of random attribute noise into the training data for a neural network was equivalent in the infinite data limit to a form of Tikhonov regularization on the magnitude of weights. Dhifallah and Lu [2021] extended these results to arbitrary noise for random feature models, which are a restricted class of neural networks. These papers supplement work designing loss functions robust to noisy data for training neural networks [Wang et al., 2019, Ma et al., 2020, Jin et al., 2021, Zhou et al., 2023] and greedily-grown decision trees [Wilton and Ye, 2024]. Here, we study the effect of noise on the regularization of sparse models based on 0-1 loss, such as sparse decision trees [Lin et al., 2020], and on linear models trained under exponential loss.

**Noise and SGD.** There has also been recent work focused on analyzing the behavior of the stochastic gradient descent (SGD) algorithm in the presence of artificially injected noise during the training process. More specifically, HaoChen et al. [2021] found that applying label noise at each step of SGD allows the ground truth function of a data distribution to be approximated arbitrarily well, while Gaussian parameter noise may instead lead to poor generalization. Blanc et al. [2020] showed that SGD with label noise acts as an implicit regularizer for models with training error. Damian et al. [2021] and Vivien et al. [2022] generalize this result by showing that SGD implicitly optimizes a regularized objective function under various regimes. Our work instead focuses on noisy data generation processes, independent of the algorithm used to optimize the objective.

## 3 Definitions and Notation

Consider a dataset $S = \{z_i = (x_i, y_i)\}_{i=1}^n$, where each $z_i \in \mathcal{Z} = \mathcal{X} \times \mathcal{Y}$ is drawn i.i.d. from an unknown true distribution $\mathcal{D}$. Here, $\mathcal{X} \in \mathbb{R}^{n \times p}$ is the input space, and $\mathcal{Y} \in \{-1, 1\}^n$ is the output space. Let $\mathcal{F}$ be a hypothesis space, where $f \in \mathcal{F}$ is a model mapping inputs to outputs, $f : \mathcal{X} \to \mathcal{Y}$. Define $\phi : \mathcal{Y} \times \mathcal{Y} \to \mathbb{R}^+$ as a loss function. For the misclassification error or 0-1 loss, we have $\phi(f(x), y) = \mathbb{1}_{[f(x) \neq y]}$. The true risk $L_\mathcal{D}(f)$ is the expected loss over the true distribution $\mathcal{D}$, given by $L_\mathcal{D}(f) = \mathbb{E}_{z \sim \mathcal{D}}[\phi(f(x), y)]$, and the empirical risk $\hat{L}_S(f)$ is the average loss on the dataset $S$ drawn from $\mathcal{D}$, calculated as $\hat{L}_S(f) = \frac{1}{n} \sum_{i=1}^n \phi(f(x_i), y_i)$. We denote by $R(f)$ an arbitrary regularization term with regularization parameter $\lambda \in \mathbb{R}^+$. Regularization induces simplicity in this work; for example, $R(\cdot)$ can represent the number of leaves in a decision tree, the length of a rule list, or $\ell_0$, $\ell_1$, or $\ell_2$ norms. We are interested in learning a model $f_\mathcal{D}^*$ that minimizes the true objective $Obj_\mathcal{D}(f)$ that combines risk and regularization:

$$Obj_\mathcal{D}(f) = L_\mathcal{D}(f) + \lambda R(f), \tag{1}$$

where $f_\mathcal{D}^* \in \arg\min_{f \in \mathcal{F}} Obj_\mathcal{D}(f)$. Since this model depends on an unknown distribution $\mathcal{D}$, we estimate it using the empirical risk minimizer $\hat{f}_S$, defined as: $\hat{f}_S \in \arg\min_{f \in \mathcal{F}} \widehat{Obj}_S(f)$, where $\widehat{Obj}_S(f) = \hat{L}_S(f) + \lambda R(f)$.

Following Fisher et al. [2019], Semenova et al. [2022, 2023], Xin et al. [2022], we define the *true Rashomon set* $R_{set_\mathcal{D}}(\mathcal{F}, \theta)$ to be

$$R_{set_\mathcal{D}}(\mathcal{F}, \theta) := \{f \in \mathcal{F} : Obj_\mathcal{D}(f) \leq Obj_\mathcal{D}(f_\mathcal{D}^*) + \theta\}, \tag{2}$$

that is, if $L_\mathcal{D}(f) + \lambda R(f) \leq L_\mathcal{D}(f_\mathcal{D}^*) + \lambda R(f_\mathcal{D}^*) + \theta$, the model $f$ is included in the Rashomon set. $\theta \geq 0$ is the additive Rashomon parameter defined by the user. Similarly, the *empirical*

*Rashomon set* $\hat{R}_{set_S}(\mathcal{F}, \theta)$, contains models within $\theta$ of the regularized empirical risk minimizer: $\hat{R}_{set_S}(\mathcal{F}, \theta) := \{f \in \mathcal{F} : \widehat{Obj}_S(f) \leq \widehat{Obj}_S(\hat{f}_S) + \theta\}$. Past work has shown that the true and empirical Rashomon sets may be similar [Semenova et al., 2022, Donnelly et al., 2023]. We will omit "true" or "empirical" and use only "Rashomon set" when it is not significant over which distribution the set is computed.

We will model noise in the labels of data with a uniform label noise model, where each label is flipped independently with the fixed probability $\rho \in \left(0, \frac{1}{2}\right)$. To sample data with random label noise from the distribution $\mathcal{D}$, we sample $z = (x, y) \sim \mathcal{D}$, then with probability $\rho$ we change the label of $y$. We denote the noisy version of this data distribution as $\mathcal{D}_\rho$. By this definition, for $x, y \sim \mathcal{D}$, with probability of $y = 1|x$ denoted as $p_y$, when sampling from $\mathcal{D}_\rho$, we have $p_y(1 - 2\rho) + \rho$ [Semenova et al., 2023]. For a finite dataset, we denote $S_\rho$ to be a dataset sampled according to distribution $\mathcal{D}_\rho$. Let $\mathcal{D}_\rho^n$ be the distribution of datasets $S_\rho$ under this noise model. As shorthand notation, define $\mathbb{E}_{S_\rho}$ to mean $\mathbb{E}_{S_\rho \sim \mathcal{D}_\rho^n}$. We assume that in practice we receive noisy data $S_\rho$ and not cleaner data $S$. (Here, $S$ does not have the uniform random label noise, but it is not necessarily clean in other ways.)

In this work, we measure how noise impacts the simplicity of the best model in the hypothesis space as well as the models in the Rashomon set. First, in Section 4, we consider random label noise and 0-1 loss, and then examine additive attribute noise for the exponential loss in Section 6. For both of these cases, we show that with more noise, we can expect simpler models.

# 4   Random Label Noise and Regularized 0-1 Loss

Noisy labels are common in real-world datasets, especially in high-stakes decision domains. There are many sources of this noise, including subjective judgments, typographical and clerical errors, and systematic biases. Next, we show that when there is noise in the labels, the regularization of the optimal model is implicitly stronger.

## 4.1   Noise Increases Regularization

We study the effect of random label noise on the optimal models for 0-1 misclassification loss. We first show that optimizing over the noisy data distribution is equivalent to optimizing over the cleaner data distribution with stronger regularization. Formally:

**Theorem 1** (Regularized 0-1 loss under random label noise)**.** *Consider true data distribution $\mathcal{D}$, and uniform label noise with noise parameter $\rho \in (0, 1/2)$. Let $\mathcal{D}_\rho$ denote the noisy version of $\mathcal{D}$. Consider 0-1 loss $L$ and let $R : \mathcal{F} \to \mathbb{R}$ be a regularization function with $\lambda \in \mathbb{R}^+$ a regularization parameter. Formally,*

$$\arg\min_{f \in \mathcal{F}} L_{\mathcal{D}_\rho}(f) + \lambda R(f) = \arg\min_{f \in \mathcal{F}} L_{\mathcal{D}}(f) + \frac{\lambda}{1 - 2\rho} R(f).$$

*Similarly, given a dataset $S$ sampled according to $\mathcal{D}$, and $S_\rho$ the noisy version of $S$, $\arg\min_{f \in \mathcal{F}} \mathbb{E}_{S_\rho} \hat{L}_{S_\rho}(f) + \lambda R(f) = \arg\min_{f \in \mathcal{F}} \hat{L}_S(f) + \frac{\lambda}{1-2\rho} R(f)$.*

Theorem 1 applies to any model class and any regularization function over the models in the model class. This includes model classes such as sparse decision trees, which regularize the number of leaves [Lin et al., 2020], rule lists, which regularize the number of rules [Angelino et al., 2017], and scoring systems, which regularize the $\ell_0$-norm of the parameter vector [Ustun and Rudin, 2016]. We prove Theorem 1 in Appendix A. This result proves the first row of Table 1, as each of these hypothesis spaces optimize 0-1 loss with a hypothesis-space-specific regularization function.

A consequence of Theorem 1, intuitively, is that the optimal model with a higher regularization penalty should be simpler and fit the original data less precisely. Formally,

**Theorem 2** (Optimal model simplifies under random label noise)**.** *Under the same assumptions as in Theorem 1, let $f_{\mathcal{D}}^*$ be the optimal model in $\mathcal{F}$ over distribution $\mathcal{D}$ and let $f_{\mathcal{D}_\rho}^*$ be the optimal model in $\mathcal{F}$ over $\mathcal{D}_\rho$. Then either $R(f_{\mathcal{D}_\rho}^*) = R(f_{\mathcal{D}}^*)$ and $L_{\mathcal{D}}(f_{\mathcal{D}_\rho}^*) = L_{\mathcal{D}}(f_{\mathcal{D}}^*)$ (same complexity model) or*

$$R(f_{\mathcal{D}_\rho}^*) < R(f_{\mathcal{D}}^*) \text{ and } L_{\mathcal{D}}(f_{\mathcal{D}_\rho}^*) > L_{\mathcal{D}}(f_{\mathcal{D}}^*) \text{ (strictly simpler model)}.$$

*An identical result applies for finite data when $f_{S_\rho}^*$ is optimized over the loss function $\mathbb{E}_{S_\rho} \hat{L}_{S_\rho}(f)$.*

A proof of Theorem 2 is in Appendix B. Corollary 10 in Appendix B gives a bound for how much simpler the noisy optimal model will be, based on its performance on the cleaner training data.

## 4.2 Noise Simplifies the Rashomon Set

In the previous section, we considered optimizing for one model, namely the best-performing model in the class. However, the Rashomon set of models provides a lot of benefits, including deeper insights into the data, flexibility in model selection by choosing a more fair, robust model or model that obeys better domain constraints, and quantification of prediction uncertainty over this set of well-performing models [Rudin et al., 2024]. Therefore, we further study how label noise influences the complexity of the models in the true Rashomon set. More specifically, we show that the models in the true Rashomon set arising from a noisy distribution cannot be more complex than those in the Rashomon set for the corresponding cleaner distribution.

Consider two true Rashomon sets $R_{set_{\mathcal{D}}}(\mathcal{F}, \theta)$ and $R_{set_{\mathcal{D}_\rho}}(\mathcal{F}, \theta)$ over cleaner data distribution $\mathcal{D}$ and noisier data distribution $\mathcal{D}_\rho$. We may partition these two sets of models into three disjoint sets: $\mathcal{F}_{both} = \{f \in \mathcal{F} : f \in R_{set_{\mathcal{D}}}(\mathcal{F}, \theta) \cap R_{set_{\mathcal{D}_\rho}}(\mathcal{F}, \theta)\}$ (the set of models in both the cleaner and noisier Rashomon sets), $\mathcal{F}_{out} = \{f \in \mathcal{F} : f \in R_{set_{\mathcal{D}}}(\mathcal{F}, \theta) \setminus R_{set_{\mathcal{D}_\rho}}(\mathcal{F}, \theta)\}$ (the set of models in the Rashomon set over the cleaner data distribution, but not the noisy one), and $\mathcal{F}_{in} = \{f \in \mathcal{F} : f \in R_{set_{\mathcal{D}_\rho}}(\mathcal{F}, \theta) \setminus R_{set_{\mathcal{D}}}(\mathcal{F}, \theta)\}$ (the set of models in the Rashomon set over the noisy data distribution, but not the cleaner one).

We are interested in studying the relationship between the complexity of models in Rashomon sets over cleaner and noisy data. In this direction, we show that under mild assumptions, any models that are in the Rashomon set over the noisier data distribution, but not the cleaner one ($\mathcal{F}_{in}$), will be simpler than the optimal model over the cleaner data. Since models in $\mathcal{F}_{both}$ have the same complexity in both Rashomon sets, and models in $\mathcal{F}_{out}$ tend to be complex (see Theorem 11 in Appendix C), this result shows that the Rashomon set over noisy data will tend to contain lower complexity models than the Rashomon set over cleaner data. Formally,

**Theorem 3** (Models that enter the noisier true Rashomon set are simple). *Consider true data distribution $\mathcal{D}$, 0-1 loss function, regularization $R(\cdot)$ and regularization parameter $\lambda$. Consider also uniform label noise, where each label is flipped independently with probability $\rho \in (0, \frac{1}{2})$. Let $\mathcal{D}_\rho$ be the noisier data distribution. If $Obj_{\mathcal{D}_\rho}(f_{\mathcal{D}}^*) > Obj_{\mathcal{D}_\rho}(f_{\mathcal{D}_\rho}^*) + 2\rho\theta$, i.e., the optimal model over the cleaner data distribution $\mathcal{D}$ is not in the Rashomon set of the noisy distribution with Rashomon parameter $2\rho\theta$, then every model from $\mathcal{F}_{in}$ in the noisier true Rashomon set $R_{set_{\mathcal{D}_\rho}}(\mathcal{F}, \theta)$ is simpler than $f_{\mathcal{D}}^*$:*

$$\forall \tilde{f} \in \mathcal{F}_{in} : R(\tilde{f}) < R(f_{\mathcal{D}}^*).$$

*More specifically, $R(\tilde{f}) < R(f_{\mathcal{D}}^*) - \frac{1}{\lambda}\left(\frac{\Delta}{2\rho} - \theta\right)$, where $\Delta = Obj_{\mathcal{D}_\rho}(f_{\mathcal{D}}^*) - Obj_{\mathcal{D}_\rho}(f_{\mathcal{D}_\rho}^*)$. Note that $\frac{\Delta}{2\rho} - \theta > 0$. An identical result applies for finite data when models are optimized over $\mathbb{E}_{S_\rho}\hat{L}_{S_\rho}(f)$.*

The proof of Theorem 3 is in Appendix C. Note that we showed in the previous section that an optimal model over the noisier data distribution tends to be simpler than an optimal model of the cleaner data distribution, $R(f_{\mathcal{D}_\rho}^*) < R(f_{\mathcal{D}}^*)$. Therefore, we believe the assumption that $f_{\mathcal{D}}^*$ is not in the noisier true Rashomon set with the Rashomon parameter $2\rho\theta$ is plausible in practical noisy settings. For smaller amounts of noise, the cleaner and noisier Rashomon sets may be similar enough to violate the assumption in Theorem 3; in this case, we expect the two Rashomon sets to overlap a lot, leading to larger $\mathcal{F}_{both}$ and similar model complexity between the Rashomon sets.

When the data is noisy, practitioners can expect to find simple-and-accurate models within the Rashomon sets for regularized 0-1 loss. We experimentally support our results in Sections 4.1 and 4.2 for empirical datasets and the expected empirical Rashomon set in Section 7 and Appendix I.

## 5  Unregularized Decision Trees and the Set of Grounded Models

The Rashomon ratio measures the size of the Rashomon set relative to the size of the hypothesis space [Semenova et al., 2022, 2023, Rudin et al., 2022]. For regularized 0-1 loss, we demonstrated that random label noise is equivalent to an increase in the regularization parameter. A larger regularization parameter penalizes more complicated models and effectively shrinks the hypothesis space, which tends to increase the Rashomon ratio [Semenova et al., 2023]. In turn, larger Rashomon ratios correspond to a higher probability of obtaining a desired (e.g., interpretable or simpler) model and correlate with the existence of simpler-yet-accurate models [Semenova et al., 2022].

However, what happens if there is no regularization in the first place and the hypothesis space does not change in size (for example, consider fully-grown decision trees without penalties on the number of leaves)? In this scenario, in the presence of noise, we show that practitioners can still expect to find simple models within the Rashomon set. However, the reason is not due to noise affecting regularization, but rather because noise makes the Rashomon set increase in size. We give theoretical evidence for this claim in Section 5.2, where we show that the *set of grounded models* (as defined in Section 5.2) grows with noise. This set usually contains the Rashomon set (see Appendix I.3). We begin with an observation about features in our dataset: noise distorts signal in high-quality features faster than in lower-quality features. This leads to an increase in the size of the *set of (relatively) good features*, which we define formally next.

## 5.1 The Set of Good Features Increases under Noise

For a dataset $S = X \times Y$, let $\mathcal{G} = \{g_j\}_{j=1}^p$ denote the set of features, where each $g_j = \{x_{\cdot,j}\}$ is the $j^{th}$ column of the feature matrix $X$. Note that $X$ can be continuous or binary. For every feature $g$, we can evaluate its quality based on how close it is to the label vector $Y$ according to a similarity function $\mathcal{M}_S(g) = \mathcal{M}(g, Y)$. Different metrics can be used as $\mathcal{M}(\cdot)$, including area under the receiver operating characteristic (ROC) curve (AUC), normalized Hamming similarity (one minus the normalized Hamming distance, which is the normalized count of different element values between two binary vectors) if $g_j$ are binary ($g_j \in \{-1, 1\}$), and correlation if both the labels and the feature are continuous. Given $\mathcal{M}_S(g)$, we define a set of good features as follows:

**Definition 4** (Set of good features). *Assume we are given a dataset $S = X \times Y$, set of features $\mathcal{G} = \{g_j\}_{j=1}^p = \{x_{\cdot,j}\}_{j=1}^p$, a feature quality metric $\mathcal{M}_S(g)$ and a parameter $\gamma$. Let $\hat{g} := \arg\max_{g \in \mathcal{G}} \mathcal{M}_S(g)$. Then we define the set of good features $G_{\mathcal{M}_S}(\mathcal{G}, \gamma)$ to be*

$$G_{\mathcal{M}_S}(\mathcal{G}, \gamma) := \{g \in \mathcal{G} : \mathcal{M}_S(g) \geq \mathcal{M}_S(\hat{g}) - \gamma\}.$$

We can think of the set of good features similarly to the Rashomon set, where the former contains all relatively good features based on the quality metric, and the Rashomon set contains all relatively good models (combinations of features) with respect to risk.

Intuitively, we expect that datasets originating from less noisy data generation processes will have higher quality features. For example, if there exists a feature with a very high AUC, then the accuracy of learned models utilizing this feature will also be high. In the presence of label noise, we can precisely calculate how the quality of each feature changes for specific cases, including unnormalized AUC (as demonstrated in Theorem 5 below, proven in Appendix D) and normalized Hamming similarity with binary features (direct consequence of proof in Theorem 1). For a balanced dataset $S$, where the number of positive and negative samples are the same and equal to $n/2$, we define the unnormalized AUC as $\overline{AUC}_S(g) = \frac{n^2}{4} AUC_S(g)$ (AUC between $g$ and the label on $S$).

**Theorem 5** (Unnormalized AUC for continuous features increases with label noise). *Consider a balanced dataset $S = X \times Y$, i.e. $\Pr(y = 1) = \Pr(y = -1)$. Let $g = x_{\cdot,j}$ be a continuous feature with distinct values $g^1 < \ldots < g^n$. Let $\overline{AUC}_S(g)$ denote the unnormalized AUC value of $g$ on the labels $Y$. Consider uniform label noise, where each label is flipped independently with probability $\rho < \frac{1}{2}$. Let $S_\rho$ be a noisier dataset. Then for every feature $g \in \mathcal{G}$:*

$$\mathbb{E}_{S_\rho}[\overline{AUC}_{S_\rho}(g)] = (1 - 2\rho)\overline{AUC}_S(g) + C(\rho, n),$$

*where $C(\rho, n) = \rho\left(\frac{n}{2}\right)\left(\frac{n}{2} + \rho - 1\right)$ is constant for a given $\rho$ and $n$.*

An important corollary directly follows from Theorem 5, which states that under noise, good features with higher AUC lose signal faster than features with lower AUC.

**Corollary 6.** *Under the same amount of uniform random label noise $\rho$, the expected unnormalized AUC of features with higher initial value decreases faster than the expected unnormalized AUC of features with lower initial value. For two features $g_1, g_2$, if $\overline{AUC}_S(g_1) < \overline{AUC}_S(g_2)$, then*

$$\overline{AUC}_S(g_1) - \mathbb{E}_{S_\rho} \overline{AUC}_{S_\rho}(g_1) < \overline{AUC}_S(g_2) - \mathbb{E}_{S_\rho} \overline{AUC}_{S_\rho}(g_2).$$

The different rate of change of features with different values of AUC also means that under noise, the set of good features increases. Since the quality metric $\mathcal{M}_{\mathbb{E}_{S_{\rho_1}}}(g)$ decreases with noise for a given feature $g$, this implies that in the noisier dataset there are more features with equivalently weak signals as compared to a cleaner dataset.

**Corollary 7.** *Consider a dataset $S = X \times Y$. Let $\{\overline{AUC}_S(g_j)\}_{j=1}^p$ be in decreasing order and spaced by distances at most $\delta$, meaning that $0 \leq \overline{AUC}_S(g_j) - \overline{AUC}_S(g_{j+1}) \leq \delta$ for each $j = 1, \ldots, p - 1$. Assume that we apply uniform label noise with flip probabilities $\rho_1$ and $\rho_2$ to $S$ to obtain $S_{\rho_1}$ and $S_{\rho_2}$, and that $|G_{\mathbb{E}_{S_{\rho_1}} \overline{AUC}_{S_{\rho_1}}}(\mathcal{G}, \gamma)| < p$. If $\rho_2 \geq \nu(\rho_1) := \frac{1}{2}\left(1 - \frac{\gamma(1-2\rho_1)}{\gamma + \delta(1-2\rho_1)}\right)$ noting that $\nu(\rho_1) > \rho_1$, then the size of the set of features which are good in expectation is strictly larger with more noise,*

$$|G_{\mathbb{E}_{S_{\rho_1}} \overline{AUC}_{S_{\rho_1}}}(\mathcal{G}, \gamma)| < |G_{\mathbb{E}_{S_{\rho_2}} \overline{AUC}_{S_{\rho_2}}}(\mathcal{G}, \gamma)|.$$

The proof of Corollary 7 is in Appendix E and more experimental results are in Appendix I.3. Note that Corollaries 6 and 7 apply as well to the case of normalized Hamming similarity.

Interestingly, Corollary 7 provides one possible explanation for the existence of large Rashomon sets. If there are more features that can explain the labels approximately-equally-well, then multiple good models could be composed of these features, as we will discuss next.

## 5.2   The Fraction of Grounded Models Increases for Unregularized Decision Trees

Consider a dataset with binary features and a hypothesis space $\mathcal{F}_d$ of fully grown decision trees of depth $d$. For example, a fully grown tree of depth 2 has three (internal) nodes (root and two child nodes) and four leaves (two leaves for each child node). Let the *set of grounded models* (set of models that use good features), $\mathcal{H}_{set_S}(\mathcal{F}_d)$, consist of models that (1) use at least one feature from the set of good features $G_{set_S}(\mathcal{G}, \gamma)$, (2) use labels determined by a majority vote of data in the leaves. Let $\mathcal{H}_{ratio_S}(\mathcal{F}_d)$ be the fraction of such models in the hypothesis space, meaning that, similar to the Rashomon ratio, $\mathcal{H}_{ratio_S}(\mathcal{F}_d) = \frac{|\mathcal{H}_{set_S}(\mathcal{F}_d)|}{|\mathcal{F}_d|}$. Often, the set of grounded models contains most of the Rashomon set, because the trees in the Rashomon set usually rely on at least one feature with a strong relationship with the label (measured by AUC; see Appendix I.3).

Next, we formally show that the fraction of grounded models increases with more random label noise. We again use the set of good features discussed above, with features ordered by AUC values.

**Theorem 8** (Fraction of grounded models increases when the set of good features increases)**.** *For a dataset $S = X \times Y$ with binary feature matrix $X \in \{-1, 1\}^{n \times p}$, consider a hypothesis space $\mathcal{F}_d$ of fully grown trees of depth $d$. Consider uniform random label noise with noise parameter $\rho$. Let $\mathcal{H}_{set_{\mathbb{E}_{S_\rho}}}(\mathcal{F}_d)$ denote the set of grounded models, based on the set of good features $G_{\mathbb{E}_{S_\rho} \overline{AUC}_{S_\rho}}(\mathcal{G}, \gamma)$. Under the assumptions of Corollary 7 on the set of good features, the fraction of grounded models increases with uniform random label noise. More formally, for $\rho_2 \geq \nu(\rho_1)$,*

$$\left|\mathcal{H}_{set_{\mathbb{E}_{S_{\rho_1}}}}(\mathcal{F}_d)\right| < \left|\mathcal{H}_{set_{\mathbb{E}_{S_{\rho_2}}}}(\mathcal{F}_d)\right| \text{ and } \mathcal{H}_{ratio_{\mathbb{E}_{S_{\rho_1}}}}(\mathcal{F}_d) < \mathcal{H}_{ratio_{\mathbb{E}_{S_{\rho_2}}}}(\mathcal{F}_d).$$

The proof of Theorem 8 is in Appendix F. The key observation made in our proof of Theorem 8 is that the set of grounded models increases in size as the set of good features grows. Since the set of grounded models typically contains the Rashomon set, this tends to increase the Rashomon ratio in unregularized hypothesis spaces. As a reminder, larger Rashomon ratios are associated with the existence of simpler-yet-accurate models. We expect the results of this section to hold for other hypothesis spaces as well, e.g., tree ensembles. So far we have considered 0-1 loss with random label noise. Unsurprisingly, we can expect simpler, more regularized, models for other losses and noise models as well. In particular, we next demonstrate that additive attribute noise acts as an implicit regularizer for the hypothesis space of linear models optimized for exponential loss.

## 6   Additive Attribute Noise and the Exponential Loss

It has been known since the 1990s that additive attribute noise to a dataset in the setting of ridge regression acts as an implicit regularizer [Bishop, 1995, Semenova et al., 2023]. More specifically, adding noise $\epsilon_i \sim \mathcal{N}(\bar{0}, \sigma^2 I)$, where $\bar{0}$ is a zero vector and $I$ is identity matrix, to every sample $x_i$ (and thus creating a new sample $x_i' = x_i + \epsilon$) implicitly increases the $\ell_2$ regularization parameter from $C$ to $C + \sigma^2$. Semenova et al. [2023] have also shown that using additive attribute noise increases the Rashomon ratio for ridge regression. However, this work does not address other continuous losses for classification. In this section, we prove that for exponential loss and binary

classification, additive attribute noise similarly functions as implicit $\ell_2$ regularization on the logarithm of the loss for an otherwise unregularized setting. We give an explicit characterization of the regularization parameter based on the variance of the noise added.

**Theorem 9** (Exponential loss under additive attribute noise). *Consider the dataset $S$ and a hypothesis space $\mathcal{F}$ of linear models, $\mathcal{F} = \{f = \omega^T x, \omega \in \mathbb{R}^p\}$. For a given model $f \in \mathcal{F}$, consider the exponential loss $L_S(f) = \frac{1}{n} \sum_{i=1}^{n} e^{-y_i \omega^T x_i}$. Let $\epsilon_i$, such that $\epsilon_i \sim \mathcal{N}(\bar{0}, \sigma^2 I)$ ($\sigma > 0$, $I$ is identity matrix), be i.i.d. noise vectors added to every sample: $x'_i = x_i + \epsilon_i$. If $\mathbb{E}_\epsilon L_{S_{\varepsilon(\sigma)}}(f)$ is the expected exponential loss under additive Gaussian noise, then*

$$\mathbb{E}_\epsilon L_{S_{\varepsilon(\sigma)}}(f) = L_S(f) \cdot e^{\frac{\sigma^2}{2} \|\omega\|_2^2},$$

*where for simplicity we denote $\mathbb{E}_{\epsilon_1,...,\epsilon_n \sim \mathcal{N}(\bar{0},\sigma^2 I)}$ as $\mathbb{E}_\epsilon$.*

A proof of Theorem 9 is in Appendix G. Immediately from Theorem 9, we have that

$$\text{argmin}_{f \in \mathcal{F}} \mathbb{E}_\varepsilon L_{S_\sigma}(f) = \text{argmin}_{f \in \mathcal{F}} \left( \log L_S(f) + \frac{\sigma^2}{2} \|\omega\|_2^2 \right).$$

In other words, additive noise introduces $\ell_2$ regularization on the logarithm of the exponential loss. Furthermore, the $\ell_2$ regularization parameter is explicitly given as $\frac{\sigma^2}{2}$. This shows that additive attribute noise encourages linear models to become simpler when there is more noise present.

# 7 Experimental Results

We now present experimental results supporting the results in Section 4 for uniform label noise and 0-1 loss and Section 6 for additive attribute noise and exponential loss. We give evidence for our finding that noise acts as an implicit regularizer and that the optimal model optimized over data with injected noise is simpler than the optimal model without additional noise. We focus our experiments in this section on criminal recidivism and financial datasets to emphasize the applicability of our work to high-stakes domains with human data. Please see Appendix I for additional experiments.

## 7.1 Sparse Decision Trees, 0-1 Loss, Random Label Noise.

We used the GOSDT-guesses algorithm by McTavish et al. [2022] to optimize sparse decision trees over varying amounts of label noise (between 0.0 and 0.3). In order to correctly simulate the results in Theorem 1, the experiment estimates the size of a *model* optimized over the *expectation* over

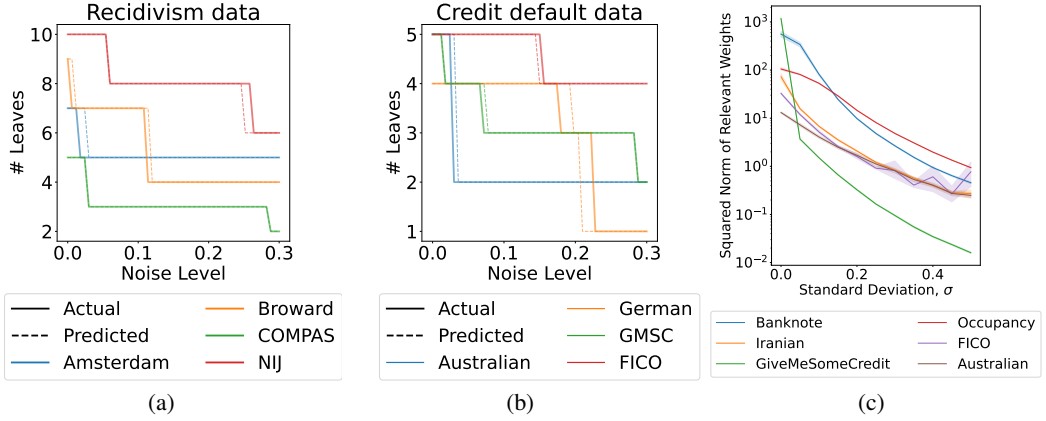

Figure 1: (a), (b): For the hypothesis space of sparse decision trees and 0-1 loss, the number of leaves in optimal models for several datasets decreases with increased label noise. The solid lines depict the observed number of leaves in an optimal model over noisy data. The dashed lines depict the number of leaves of the optimal model over the cleaner data with regularization $\frac{\lambda}{1-2\rho}$ (see Theorem 1). (c): For the hypothesis space of linear models and exponential loss, the sum of the squares of the weights corresponding to the continuous features decreases as additive noise with standard deviation $\sigma$ is applied to the dataset.

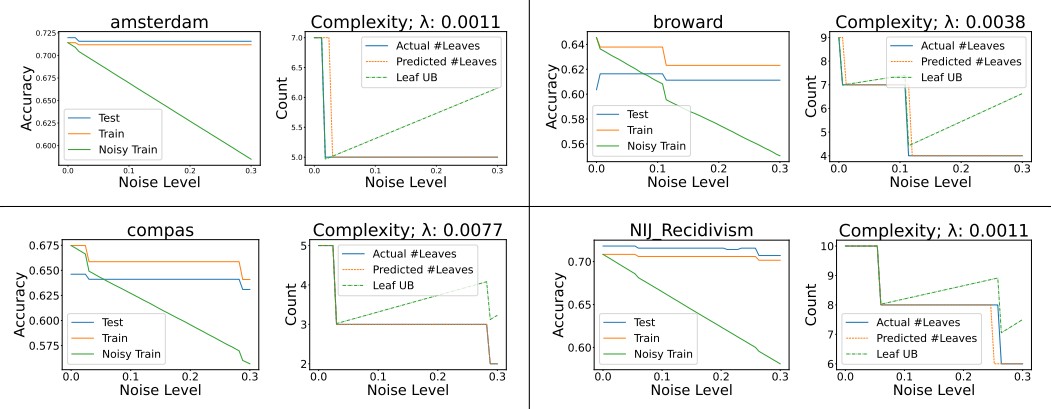

Figure 2: Experimental results for Section 4 on recidivism datasets. In accuracy plots (left), blue is accuracy on cleaner test data, orange is accuracy on cleaner train data, and green is accuracy on noisy train data. In complexity plots, blue corresponds to leaves in optimal model over noisy data, orange to optimal model over cleaner data with higher regularization as in Figure 1. Green is the leaf upper bound from Corollary 10. Lambda is regularization parameter optimized via 5-fold CV.

noise draws of the loss (i.e., $R(f_{S_\rho}^*)$), where $f_{S_\rho}^* \in \arg\min_{f \in \mathcal{F}} \mathbb{E}_{S_\rho} \hat{L}_{S_\rho} + \lambda R(f)$). This is different from taking the *expectation* of the *size* of models optimized over data with only a single noise draw ($\mathbb{E}_{S_\rho} R(f^*)$, where $f^* \in \arg\min_{f \in \mathcal{F}} \hat{L}_{S_\rho}(f) + \lambda R(f)$). To approximate optimization over expected noise draws, we concatenated 250 noise draws into a single dataset upon which to optimize a decision tree. The full experimental design is presented in Appendix I. The results of this experiment are shown in Figure 1(a)-(b). These results demonstrate simplification in accordance with Theorem 2 and the alignment between models trained on cleaner data with varying regularization and models trained on noisy data with consistent regularization (Theorem 1).

An interesting observation from these experiments is that as we increase the label noise parameter, the generalization gap between the accuracy on the cleaner train and test set tends to shrink and the test accuracy remains very stable (see Figures 2, 6, 7). This is what we would expect given Theorems 1 and 2, since optimizing over the expectation of noise increases the effective regularization in the cleaner problem.

### 7.2 Linear Models, Exponential Loss, Additive Attribute Noise

To show that additive attribute noise has a regularizing effect on linear models under the exponential loss, we computed the optimal models on datasets under different noise levels and compared their complexities, measured by the norm of the weights. For each dataset, we sampled 100 independent noise draws for 10 different noise levels with $\sigma \in [0.05, 0.5]$. For each noise draw, we computed the optimal linear model on the noisy dataset using gradient descent. In Figure 1(c), we observe that, as the noise level $\sigma$ increases, the complexity of the optimal model rapidly decreases. This corroborates the regularizing effect of noise demonstrated in Theorem 9.

### 7.3 Empirical Evidence that Rashomon Sets Over Noisier Data Contain Simpler Models

In Figure 3, we show empirical evidence that the complexity of the Rashomon set of sparse decision trees tends to decrease with the injection of label noise. Moreover, the more noise added to the dataset, the simpler the models in the Rashomon set become. We provide detailed descriptions of the experiments in this section in Appendix I.

## 8 Limitations and Future Work

One limitation of our work is that the theoretical results in Section 4 apply to models optimized over the expected loss over the distribution of possible noise draws with label-flip probability $\rho$. For larger sample sizes, we expect optimizing over the expectation of noise draws to behave similarly to optimizing over a single noise draw. However, for smaller sample sizes, the optimal model optimized over a single noise draw may deviate from the optimal model over the expectation of noise draws.

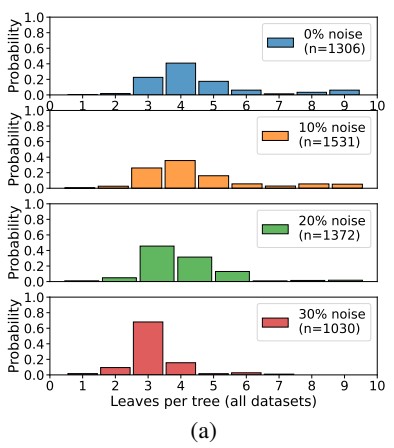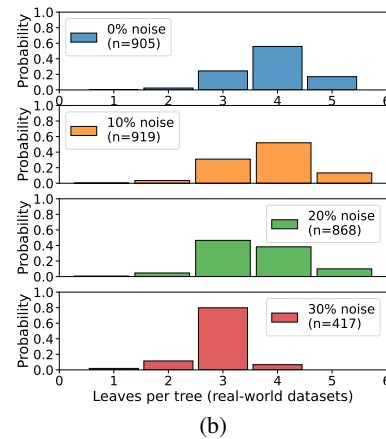

(a)          (b)

Figure 3: A visual demonstration of the simplifying effect of noise on Rashomon sets. This shows a bar chart of the discrete probability distribution of the number of leaves among models in the Rashomon set. Results are shown aggregated in (a) for 23 real-world and synthetic datasets, and in (b), just real-world data, for 9 recidivism and finance datasets.

One possible future direction is to bound the expected complexity of a model optimized over a single noise draw in terms of sample size and the probability of flipping labels.

A natural extension of our work is to adapt our results to other losses, like hinge loss and logistic loss. These results are not immediate, because in loss functions more complicated than 0-1 loss, the error considers distances to the decision boundary. Focusing on specific hypothesis spaces rather than on particular choices of loss function (e.g., rule lists optimized on logistic loss, or GAMs optimized on exponential loss) can also help to produce more specific and possibly tighter results similar to those as in Section 6.

We can also try to generalize to other noise models (i.e. random flipping noise in the inputs, or constrained non-uniform noise in the outputs). Dropping the assumption of uniform random label noise needs other techniques as opposed to those that are used in Theorem 1, as the optimization problems are no longer equivalent. If there is non-uniform noise *at least* $\rho$, it can be decomposed into the non-uniform distribution plus $\rho$ uniform label noise, and our results in Theorem 1 can be applied. However, such decomposition is not always realistic as, for example, it is very likely that some features are not noisy.

## 9    Practical Guidance for High-Stakes Decision Domains

The fallacy of the premise of the movie Minority Report is that it is possible to predict with perfect accuracy whether someone will commit a crime in the future. In reality, predictions of recidivism are made ∼2 or 3 years in advance, giving time for a multitude of random interactions in the world to take place. This unpredictability leads, as we showed in this paper, to inherent regularization, and provably simpler models than if noise were not present. Rather than assuming that increased algorithmic sophistication in the future can potentially lead to improved accuracy using the same types of data, it is more realistic to assume that the distributions of data in the future are generally similar to those in the present, and that our best ML methods already reach an approximate performance maximum for these types of data. This latter view clears the way for policy-makers to regulate the use of simpler models. Already the use of black box models has led to individuals being denied freedom based on typographical errors [Wexler, 2017a,b, Rudin et al., 2020] and patients being deceived about the value of expensive medical treatment options [Afnan et al., 2021].

Our findings underscore the critical importance of using simpler models for datasets affected by noise, thereby prioritizing model interpretability and transparency. We believe that this understanding can empower policymakers to advocate for the use of simple, interpretable models, ensuring the trustworthy, accessible, and equitable deployment of AI systems in high-stakes decision domains.

## Acknowledgments and Disclosure of Funding

We acknowledge funding from the Department of Energy under grant DE-SC0023194, the National Institutes of Health under grant 5R01-DA054994, and a grant from Fujitsu.

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

# A  Proof of Theorem 1

**Theorem 1** (Regularized 0-1 Loss under Random Label Noise). *Consider true data distribution $\mathcal{D}$, and uniform label noise with noise parameter $\rho \in (0, 1/2)$. Let $\mathcal{D}_\rho$ denote the noisy version of $\mathcal{D}$. Consider 0-1 loss $L$ and let $R : \mathcal{F} \to \mathbb{R}$ be a regularization function with $\lambda \in \mathbb{R}^+$ a regularization parameter. Formally,*

$$\arg \min_{f \in \mathcal{F}} L_{\mathcal{D}_\rho}(f) + \lambda R(f) = \arg \min_{f \in \mathcal{F}} L_{\mathcal{D}}(f) + \frac{\lambda}{1 - 2\rho} R(f).$$

*Similarly, given a dataset $S$ sampled according to $\mathcal{D}$, and $S_\rho$ the noisy version of $S$, $\arg \min_{f \in \mathcal{F}} \mathbb{E}_{S_\rho} \hat{L}_{S_\rho}(f) + \lambda R(f) = \arg \min_{f \in \mathcal{F}} \hat{L}_S(f) + \frac{\lambda}{1-2\rho} R(f)$.*

*Proof.* Recall that the true risk for 0-1 loss $L_{\mathcal{D}}(f) = \mathbb{E}_{z=(x,y)\sim\mathcal{D}}[l(f, z)] = \mathbb{E}_{(x,y)\sim\mathcal{D}}[\mathbb{1}_{[f(x)\neq y]}]$. Semenova et al. [2023] show that $L_{\mathcal{D}_\rho}(f) = (1 - 2\rho)L_{\mathcal{D}}(f) + \rho$. This is independent of regularization, so we have

$$L_{\mathcal{D}_\rho}(f) + \lambda R(f) = (1 - 2\rho)L_{\mathcal{D}}(f) + \lambda R(f) + \rho.$$

Since the minimization procedure is invariant to shifts and scales by constants, we conclude

$$\arg \min_{f \in \mathcal{F}} L_{\mathcal{D}_\rho}(f) + \lambda R(f) = \arg \min_{f \in \mathcal{F}} (1 - 2\rho)L_{\mathcal{D}}(f) + \lambda R(f) + \rho$$

$$= \arg \min_{f \in \mathcal{F}} (1 - 2\rho)L_{\mathcal{D}}(f) + \lambda R(f)$$

$$= \arg \min_{f \in \mathcal{F}} L_{\mathcal{D}}(f) + \frac{\lambda}{1 - 2\rho} R(f).$$

An identical argument applies to the finite data case, where we use the remark in the proof of Theorem 2 of [Semenova et al., 2023]. $\square$

# B  Proof of Theorem 2

**Theorem 2** (Optimal model simplifies under random label noise). *Under the same assumptions as in Theorem 1, let $f_{\mathcal{D}}^*$ be the optimal model in $\mathcal{F}$ over distribution $\mathcal{D}$ and let $f_{\mathcal{D}_\rho}^*$ be the optimal model in $\mathcal{F}$ over $\mathcal{D}_\rho$. Then either $R(f_{\mathcal{D}_\rho}^*) = R(f_{\mathcal{D}}^*)$ and $L_{\mathcal{D}}(f_{\mathcal{D}_\rho}^*) = L_{\mathcal{D}}(f_{\mathcal{D}}^*)$ (same complexity model) or*

$$R(f_{\mathcal{D}_\rho}^*) < R(f_{\mathcal{D}}^*) \text{ and } L_{\mathcal{D}}(f_{\mathcal{D}_\rho}^*) > L_{\mathcal{D}}(f_{\mathcal{D}}^*) \text{ (strictly simpler model)}.$$

*An identical result applies for finite data when $f_{S_\rho}^*$ is optimized over the loss function $\mathbb{E}_{S_\rho} \hat{L}_{S_\rho}(f)$.*

*Proof.* Consider optimal models

$$f_{\mathcal{D}}^* \in \arg \min_{f \in \mathcal{F}} L_{\mathcal{D}}(f) + \lambda R(f)$$

$$f_{\mathcal{D}_\rho}^* \in \arg \min_{f \in \mathcal{F}} L_{\mathcal{D}}(f) + \frac{\lambda}{1 - 2\rho} R(f). \qquad \text{(by Theorem 1)}$$

Then, by definition,

$$L_{\mathcal{D}}(f_{\mathcal{D}}^*) + \lambda R(f_{\mathcal{D}}^*) \leq L_{\mathcal{D}}(f_{\mathcal{D}_\rho}^*) + \lambda R(f_{\mathcal{D}_\rho}^*)$$

$$\text{and}$$

$$L_{\mathcal{D}}(f_{\mathcal{D}_\rho}^*) + \frac{\lambda}{1 - 2\rho} R(f_{\mathcal{D}_\rho}^*) \leq L_{\mathcal{D}}(f_{\mathcal{D}}^*) + \frac{\lambda}{1 - 2\rho} R(f_{\mathcal{D}}^*).$$

This gives

$$\lambda \left( R(f_{\mathcal{D}}^*) - R(f_{\mathcal{D}_\rho}^*) \right) \leq L_{\mathcal{D}}(f_{\mathcal{D}_\rho}^*) - L_{\mathcal{D}}(f_{\mathcal{D}}^*) \leq \frac{\lambda}{1 - 2\rho} \left( R(f_{\mathcal{D}}^*) - R(f_{\mathcal{D}_\rho}^*) \right). \qquad (3)$$

Assume for contradiction that $R(f_{\mathcal{D}_\rho}^*) > R(f_{\mathcal{D}}^*) \iff R(f_{\mathcal{D}}^*) - R(f_{\mathcal{D}_\rho}^*) < 0$. Then

$$L_{\mathcal{D}}(f_{\mathcal{D}_\rho}^*) - L_{\mathcal{D}}(f_{\mathcal{D}}^*) \geq \lambda \left( R(f_{\mathcal{D}}^*) - R(f_{\mathcal{D}_\rho}^*) \right)$$

$$\iff \frac{L_{\mathcal{D}}(f_{\mathcal{D}_\rho}^*) - L_{\mathcal{D}}(f_{\mathcal{D}}^*)}{R(f_{\mathcal{D}}^*) - R(f_{\mathcal{D}_\rho}^*)} \leq \lambda$$

and

$$L_{\mathcal{D}}(f_{\mathcal{D}_\rho}^*) - L_{\mathcal{D}}(f_{\mathcal{D}}^*) \leq \frac{\lambda}{1-2\rho} \left( R(f_{\mathcal{D}}^*) - R(f_{\mathcal{D}_\rho}^*) \right)$$

$$\iff \frac{L_{\mathcal{D}}(f_{\mathcal{D}_\rho}^*) - L_{\mathcal{D}}(f_{\mathcal{D}}^*)}{R(f_{\mathcal{D}}^*) - R(f_{\mathcal{D}_\rho}^*)} \geq \frac{\lambda}{1-2\rho}.$$

Since $\frac{\lambda}{1-2\rho} > \lambda$ for $0 < \rho < \frac{1}{2}$, this is a contradiction. Thus, $R(f_{\mathcal{D}_\rho}^*) \leq R(f_{\mathcal{D}}^*)$ which immediately gives $L_{\mathcal{D}}(f_{\mathcal{D}_\rho}^*) \geq L_{\mathcal{D}}(f_{\mathcal{D}}^*)$. The two cases in the lemma follow from (3). An identical argument holds for the finite data case. $\qquad \square$

We now highlight an intermediate step in the proof of Theorem 2, which gives us an upper bound on the complexity of an optimal model over a noisy data distribution.

**Corollary 10.** *Under the same assumptions as in Theorem 2,*

$$R(f_{\mathcal{D}_\rho}^*) \leq R(f_{\mathcal{D}}^*) - \frac{1-2\rho}{\lambda} \left( L_{\mathcal{D}}(f_{\mathcal{D}_\rho}^*) - L_{\mathcal{D}}(f_{\mathcal{D}}^*) \right).$$

*This follows immediately from (3). As we expect, the same result holds for finite sample $S$ and $S_\rho$ sampled from distributions $\mathcal{D}$ and $\mathcal{D}_\rho$.*

Note that, for a fixed amount of label noise, this bound depends only on the complexity of the optimal model over cleaner data, and the difference in loss over the cleaner distribution between optimal models optimized over noisy and cleaner data. The amount of implicit simplification due to noise depends on the degradation of generalization performance caused by noise. We plot this bound in Figures 2, 6, and 7, noting that the bound loosens as $\rho$ increases and the generalization difference remains constant.

## C  Proof of Theorem 3

**Theorem 3** (Models that enter the noisier true Rashomon set are simple)**.** *Consider true data distribution $\mathcal{D}$, 0-1 loss function, regularization $R(\cdot)$ and regularization parameter $\lambda$. Consider also uniform label noise, where each label is flipped independently with probability $\rho \in (0, \frac{1}{2})$. Let $\mathcal{D}_\rho$ be the noisier data distribution. If $Obj_{\mathcal{D}_\rho}(f_{\mathcal{D}}^*) > Obj_{\mathcal{D}_\rho}(f_{\mathcal{D}_\rho}^*) + 2\rho\theta$, i.e., the optimal model over the cleaner data distribution $\mathcal{D}$ is not in the Rashomon set of the noisy distribution with Rashomon parameter $2\rho\theta$, then every model from $\mathcal{F}_{in}$ in the noisier true Rashomon set $R_{set_{\mathcal{D}_\rho}}(\mathcal{F}, \theta)$ is simpler than $f_{\mathcal{D}}^*$:*

$$\forall \tilde{f} \in \mathcal{F}_{in} : R(\tilde{f}) < R(f_{\mathcal{D}}^*).$$

*More specifically, $R(\tilde{f}) < R(f_{\mathcal{D}}^*) - \frac{1}{\lambda} \left( \frac{\Delta}{2\rho} - \theta \right)$, where $\Delta = Obj_{\mathcal{D}_\rho}(f_{\mathcal{D}}^*) - Obj_{\mathcal{D}_\rho}(f_{\mathcal{D}_\rho}^*)$. Note that $\frac{\Delta}{2\rho} - \theta > 0$. An identical result applies for finite data when models are optimized over $\mathbb{E}_{S_\rho} \hat{L}_{S_\rho}(f)$.*

*Proof.* Consider $\tilde{f} \in \mathcal{F}_{in}$. Then we have that $\tilde{f} \notin R_{set_{\mathcal{D}}}(\mathcal{F}, \theta), \tilde{f} \in R_{set_{\mathcal{D}_\rho}}(\mathcal{F}, \theta)$, and therefore:

$$L_{\mathcal{D}}(\tilde{f}) + \lambda R(\tilde{f}) > L_{\mathcal{D}}(f_{\mathcal{D}}^*) + \lambda R(f_{\mathcal{D}}^*) + \theta, \tag{4}$$

$$L_{\mathcal{D}_\rho}(\tilde{f}) + \lambda R(\tilde{f}) \leq L_{\mathcal{D}_\rho}(f_{\mathcal{D}_\rho}^*) + \lambda R(f_{\mathcal{D}_\rho}^*) + \theta. \tag{5}$$

Additionally, from Semenova et al. [2023], we have:

$$L_{\mathcal{D}_\rho}(\tilde{f}) = (1-2\rho)L_{\mathcal{D}}(\tilde{f}) + \rho. \tag{6}$$

Thus we have:

$$
\begin{aligned}
Obj_{\mathcal{D}_\rho}(f^*_{\mathcal{D}_\rho}) + \theta =& L_{\mathcal{D}_\rho}(f^*_{\mathcal{D}_\rho}) + \lambda R(f^*_{\mathcal{D}_\rho}) + \theta \\
&\overset{(5)}{\geq} L_{\mathcal{D}_\rho}(\tilde{f}) + \lambda R(\tilde{f}) \\
&\overset{(6)}{=} (1 - 2\rho) L_{\mathcal{D}}(\tilde{f}) + \rho + \lambda R(\tilde{f}) \\
&\overset{(4)}{>} (1 - 2\rho) \left( L_{\mathcal{D}}(f^*_{\mathcal{D}}) + \lambda R(f^*_{\mathcal{D}}) + \theta - \lambda R(\tilde{f}) \right) + \rho + \lambda R(\tilde{f}) \\
&= (1 - 2\rho) Obj_{\mathcal{D}}(f^*_{\mathcal{D}}) + (1 - 2\rho)\theta + \rho + 2\rho\lambda R(\tilde{f}).
\end{aligned}
$$

Therefore, we get that

$$
R(\tilde{f}) < \frac{1}{2\rho\lambda} \left( 2\rho\theta - \rho + \left( Obj_{\mathcal{D}_\rho}(f^*_{\mathcal{D}_\rho}) - (1 - 2\rho) Obj_{\mathcal{D}}(f^*_{\mathcal{D}}) \right) \right). \tag{7}
$$

Let's focus on the difference between two objectives:

$$
\begin{aligned}
Obj_{\mathcal{D}_\rho}(f^*_{\mathcal{D}_\rho}) - (1 - 2\rho) Obj_{\mathcal{D}}(f^*_{\mathcal{D}}) &= Obj_{\mathcal{D}_\rho}(f^*_{\mathcal{D}_\rho}) - (1 - 2\rho) L_{\mathcal{D}}(f^*_{\mathcal{D}}) - (1 - 2\rho)\lambda R(f^*_{\mathcal{D}}) \\
&= Obj_{\mathcal{D}_\rho}(f^*_{\mathcal{D}_\rho}) - (1 - 2\rho) L_{\mathcal{D}}(f^*_{\mathcal{D}}) - \rho - (1 - 2\rho)\lambda R(f^*_{\mathcal{D}}) + \rho \\
&\overset{(6)}{=} Obj_{\mathcal{D}_\rho}(f^*_{\mathcal{D}_\rho}) - L_{\mathcal{D}_\rho}(f^*_{\mathcal{D}}) - \lambda R(f^*_{\mathcal{D}}) + 2\rho\lambda R(f^*_{\mathcal{D}}) + \rho \\
&= Obj_{\mathcal{D}_\rho}(f^*_{\mathcal{D}_\rho}) - Obj_{\mathcal{D}_\rho}(f^*_{\mathcal{D}}) + 2\rho\lambda R(f^*_{\mathcal{D}}) + \rho \\
&= 2\rho\lambda R(f^*_{\mathcal{D}}) + \rho - \Delta,
\end{aligned}
$$

where $\Delta = Obj_{\mathcal{D}_\rho}(f^*_{\mathcal{D}}) - Obj_{\mathcal{D}_\rho}(f^*_{\mathcal{D}_\rho})$. Since by assumption $f^*_{\mathcal{D}} \notin R_{set_{\mathcal{D}_\rho}}(\mathcal{F}, 2\rho\theta)$, then $\Delta > 2\rho\theta$ and $\frac{\Delta}{2\rho} - \theta > 0$. Therefore in (7) we have:

$$
\begin{aligned}
R(\tilde{f}) &< \frac{1}{2\rho\lambda} \left( 2\rho\theta - \rho + \left( Obj_{\mathcal{D}_\rho}(f^*_{\mathcal{D}_\rho}) - (1 - 2\rho) Obj_{\mathcal{D}}(f^*_{\mathcal{D}}) \right) \right) \\
&= \frac{1}{2\rho\lambda} \left( 2\rho\theta - \rho + 2\rho\lambda R(f^*_{\mathcal{D}}) + \rho - \Delta \right) \\
&= R(f^*_{\mathcal{D}}) - \frac{1}{\lambda} \left( \frac{\Delta}{2\rho} - \theta \right) \\
&< R(f^*_{\mathcal{D}}).
\end{aligned}
$$

An identical proof applies to the finite dataset $S$, where the model $f$ is in the noisier expected Rashomon set if $\mathbb{E}_{S_\rho} \hat{L}_{S_\rho}(f) + R(f) \leq \mathbb{E}_{S_\rho} \hat{L}_{S_\rho}(f^*_{\mathbb{E}_{S_\rho}}) + R(f^*_{\mathbb{E}_{S_\rho}})$, where $f^*_{\mathbb{E}_{S_\rho}} \in \arg\min_{f \in \mathcal{F}} \mathbb{E}_{S_\rho} \hat{L}_{S_\rho}(f) + R(f)$. $\qquad\square$

**Models in $F_{out}$ are complex.** As the theorem above states, models in $F_{in}$ in the noisy Rashomon set tend to be simpler. Now, we will show that the models in $F_{out}$ in the cleaner Rashomon set are complex models that potentially will overfit the noisier data. We illustrated this empirically in Section 7.3 and now prove it more formally in the next theorem.

**Theorem 11** (Models that exit the cleaner true Rashomon set are complex). *Consider true data distribution $\mathcal{D}$, 0-1 loss function, regularization $R(\cdot)$ and regularization parameter $\lambda$. Consider also uniform label noise, where each label is flipped independently with probability $\rho < \frac{1}{2}$. Let $\mathcal{D}_\rho$ be the noisier data distribution. If $Obj_D(f^*_{\mathcal{D}_\rho}) > Obj_D(f^*_{\mathcal{D}}) + 2\rho\theta$, i.e., the optimal model over noisy data distribution $\mathcal{D}_\rho$ is not in the cleaner true Rashomon set with Rashomon parameter $2\rho\theta$ (note that this is a symmetric assumption to the assumption in Theorem 3), then every model from $\mathcal{F}_{out}$ in the cleaner true Rashomon set $R_{set_{\mathcal{D}}}(\mathcal{F}, \theta)$ is more complex than $f^*_{\mathcal{D}_\rho}$:*

$$
\forall f \in \mathcal{F}_{out} : R(f) > R(f^*_{\mathcal{D}_\rho}) + 2(1 - \rho)\frac{\theta}{\lambda}.
$$

*Proof.* Let $f \in F_{out}$. By the definition of $F_{out}$, we have

$$
Obj_{\mathcal{D}_\rho}(f) > Obj_{\mathcal{D}_\rho}(f^*_{\mathcal{D}_\rho}) + \theta, \tag{8}
$$

and

$$Obj_{\mathcal{D}}(f) \leq Obj_{\mathcal{D}}(f_{\mathcal{D}}^*) + \theta. \tag{9}$$

From (8), (6) and by the definition of the objective, we have that,

$$(1 - 2\rho)L_{\mathcal{D}}(f) + \lambda R(f) > (1 - 2\rho)L_{\mathcal{D}}(f_{\mathcal{D}_\rho}^*) + \lambda R(f_{\mathcal{D}_\rho}^*) + \theta,$$

and

$$(1 - 2\rho)Obj_{\mathcal{D}}(f) + 2\rho\lambda R(f) > (1 - 2\rho)Obj_{\mathcal{D}}(f_{\mathcal{D}_\rho}^*) + 2\rho\lambda R(f_{\mathcal{D}_\rho}^*) + \theta.$$

We use Equation (9) to substitute in $Obj_{\mathcal{D}}(f_{\mathcal{D}}^*) + \theta$ for $Obj_{\mathcal{D}}(f)$ and obtain

$$(1 - 2\rho)(Obj_{\mathcal{D}}(f_{\mathcal{D}}^*) + \theta) + 2\rho\lambda R(f) > (1 - 2\rho)Obj_{\mathcal{D}}(f_{\mathcal{D}_\rho}^*) + 2\rho\lambda R(f_{\mathcal{D}_\rho}^*) + \theta.$$

We can rearrange this inequality to solve for the regularization $R(f)$ of the models that exit the cleaner true Rashomon set:

$$R(f) > R(f_{\mathcal{D}_\rho}^*) + \frac{\theta}{\lambda} + \frac{1 - 2\rho}{2\rho\lambda}(Obj_{\mathcal{D}}(f_{\mathcal{D}_\rho}^*) - Obj_{\mathcal{D}}(f_{\mathcal{D}}^*)).$$

By assumption, $Obj_{\mathcal{D}}(f_{\mathcal{D}_\rho}^*) > Obj_{\mathcal{D}}(f_{\mathcal{D}}^*) + 2\rho\theta$ and, therefore we get that

$$R(f) > R(f_{\mathcal{D}_\rho}^*) + \frac{\theta}{\lambda} + \frac{1 - 2\rho}{2\rho\lambda}2\rho\theta = R(f_{\mathcal{D}_\rho}^*) + \frac{2(1 - \rho)\theta}{\lambda}.$$

$\square$

# D   Proof of Theorem 5

For a given continuous feature $g$ with values $g^1 \leq \ldots \leq g^T$ and a set of binary labels $Y$ for each sample, the AUC of $g$ is defined as the area under the ROC curve of $g$, where the ROC curve is plotted against the false positive rate (FPR) and true positive rate (TPR) of a decision stump on $g$. We can choose instead to plot the absolute number of negatively and positively classified camples, forming the *unnormalized ROC curve* of $g$. The resulting *unnormalized AUC* of $g$ is defined as the area under the unnormalized ROC curve of $g$, which can be written as

$$\overline{AUC}_S(g) = n^+ n^- AUC_S(g),$$

where $n^+$ and $n^-$ are the number of positively and negatively labelled samples respectively. For more explicit notation in this section, we will use $\overline{AUC}(g, Y)$ to denote unnormalized AUC, where $\overline{AUC}(g, Y) = \overline{AUC}_S(g)$. If $g$ has $T$ distinct values, then the unnormalized AUC has a closed-form:

$$\overline{AUC}(g, Y) = \sum_{i=1}^{T} \sum_{j=i+1}^{T} \mathbb{1}_{y_i=1 \wedge y_j=-1}. \tag{10}$$

Intuitively, equation 10 counts the number of unit squares below the unnormalized ROC curve. We now recall and prove Theorem 5.

**Theorem 5.** *Consider a balanced dataset $S = X \times Y$, i.e. $\Pr(y = 1) = \Pr(y = -1)$. Let $g = x_{.,j}$ be a continuous feature with distinct values $g^1 < \ldots < g^n$. Let $\overline{AUC}_S(g)$ denote the unnormalized AUC value of $g$ on the labels $Y$. Consider uniform label noise, where each label is flipped independently with probability $\rho < \frac{1}{2}$. Let $S_\rho$ be a noisier dataset. Then for every feature $g \in \mathcal{G}$:*

$$\mathbb{E}_{S_\rho}[\overline{AUC}_{S_\rho}(g)] = (1 - 2\rho)\overline{AUC}_S(g) + C(\rho, n),$$

*where $C(\rho, n) = \rho \left(\frac{n}{2}\right) \left(\frac{n}{2} + \rho - 1\right)$ is constant for a given $\rho$ and $n$.*

*Proof.* Under the noise model, each label is flipped independently with probability $\rho$, where $1 - \rho$ is then the probability that the label was not flipped. Using equation 10, we can write

$$
\begin{aligned}
\mathbb{E}_{\tilde{Y}}[\overline{AUC}(g, \tilde{Y})] &= \mathbb{E}_{\tilde{Y}} \left[ \sum_{i<j} \mathbb{1}_{\tilde{y}_i = 1 \wedge \tilde{y}_j = -1} \right] \\
&= \sum_{i<j} \mathbb{E}_{\tilde{Y}} \left[ \mathbb{1}_{\tilde{y}_i = 1 \wedge \tilde{y}_j = -1} \right] \\
&= \sum_{i<j} \Pr(\tilde{y}_i = 1, \tilde{y}_j = -1) \\
&= \sum_{i<j} [\mathbb{1}_{y_i = -1, y_j = -1} \Pr(y_i \neq \tilde{y}_i, y_j = \tilde{y}_j) \\
&\quad + \mathbb{1}_{y_i = 1, y_j = -1} \Pr(y_i = \tilde{y}_i, y_j = \tilde{y}_j) \\
&\quad + \mathbb{1}_{y_i = -1, y_j = 1} \Pr(y_i \neq \tilde{y}_i, y_j \neq \tilde{y}_j) \\
&\quad + \mathbb{1}_{y_i = 1, y_j = 1} \Pr(y_i = \tilde{y}_i, y_j \neq \tilde{y}_j)] \\
&= \sum_{i<j} [\rho(1 - \rho) \mathbb{1}_{y_i = -1, y_j = -1} + (1 - \rho)^2 \mathbb{1}_{y_i = 1, y_j = -1} \\
&\quad + \rho^2 \mathbb{1}_{y_i = -1, y_j = 1} + \rho(1 - \rho) \mathbb{1}_{y_i = 1, y_j = 1}].
\end{aligned}
$$

Consider each of the terms separately. We have that

$$
\sum_{i<j} \mathbb{1}_{y_i = -1, y_j = -1} = \sum_{i<j} \mathbb{1}_{y_i = 1, y_j = 1} = \binom{n/2}{2},
$$

since there are $\frac{n}{2}$ positive and negative samples, so there are $\binom{n/2}{2}$ unordered pairs of each. We also have by definition that

$$
\sum_{i<j} \mathbb{1}_{y_i = 1, y_j = -1} = \overline{AUC}(g, Y),
$$

and since there are $(\frac{n}{2})^2$ unordered pairs of exactly one positive and one negative value, we lastly have that

$$
\sum_{i<j} \mathbb{1}_{y_i = -1, y_j = 1} = \left(\frac{n}{2}\right)^2 - \overline{AUC}(g, Y).
$$

Thus, putting it all together, we obtain the following for the unnormalized AUC:

$$
\begin{aligned}
\mathbb{E}_{\tilde{Y}}[\overline{AUC}(g, \tilde{Y})] &= (1 - \rho)^2 \overline{AUC}(g, Y) + \rho^2 \left( \left(\frac{n}{2}\right)^2 - \overline{AUC}(g, Y) \right) + 2\rho(1 - \rho) \binom{\frac{n}{2}}{2} \\
&= (1 - 2\rho) \overline{AUC}(g, Y) + \rho^2 \left(\frac{n}{2}\right)^2 + \rho(1 - \rho) \left(\frac{n}{2}\right) \left(\frac{n}{2} - 1\right) \\
&= (1 - 2\rho) \overline{AUC}(g, Y) + \rho \left(\frac{n}{2}\right) \left(\frac{n}{2} + \rho - 1\right).
\end{aligned}
$$

$\square$

If the dataset is not balanced, then we can use an identical proof to show that

$$
\mathbb{E}_{\tilde{Y}} \overline{AUC}(g, \tilde{Y}) = (1 - 2\rho) \overline{AUC}(g, Y) + C(\rho, n^+, n^-),
$$

where $C(\rho, n^+, n^-) = -\rho(1 - 2\rho) n^+ n^- + \rho(1 - \rho) \binom{n^+ + n^-}{2}$.

# E Proof of Corollary 7

**Corollary 7.** *Consider a dataset $S = X \times Y$. Let $\{\overline{AUC}_S(g_j)\}_{j=1}^p$ be in decreasing order and spaced by distances at most $\delta$, meaning that $0 \leq \overline{AUC}_S(g_j) - \overline{AUC}_S(g_{j+1}) \leq \delta$ for each $j = 1, \ldots, p - 1$. Assume that we apply uniform label noise with flip probabilities $\rho_1$ and $\rho_2$ to $S$ to obtain $S_{\rho_1}$ and $S_{\rho_2}$, and that $|G_{\mathbb{E}_{S_{\rho_1}} \overline{AUC}_{S_{\rho_1}}}(\mathcal{G}, \gamma)| < p$. If $\rho_2 \geq \nu(\rho_1) := \frac{1}{2}\left(1 - \frac{\gamma(1 - 2\rho_1)}{\gamma + \delta(1 - 2\rho_1)}\right)$ noting that $\nu(\rho_1) > \rho_1$, then the size of the set of features which are good in expectation is strictly larger with more noise,*

$$|G_{\mathbb{E}_{S_{\rho_1}} \overline{AUC}_{S_{\rho_1}}}(\mathcal{G}, \gamma)| < |G_{\mathbb{E}_{S_{\rho_2}} \overline{AUC}_{S_{\rho_2}}}(\mathcal{G}, \gamma)|.$$

*Proof.* Note that the features $g_1, \ldots g_p$ are in decreasing order by their unnormalized AUC, since by the corollary statement, $0 \leq \overline{AUC}_S(g_j) - \overline{AUC}_S(g_{j+1}) \leq \delta$. From Theorem 5, we know that

$$\mathbb{E}_{S_\rho} \overline{AUC}_{S_\rho}(g) = (1 - 2\rho)\overline{AUC}_S(g) + C(\rho, n),$$

where $C(\rho, n) = \rho\left(\frac{n}{2}\right)\left(\frac{n}{2} + \rho - 1\right)$ is a constant that depends only on $\rho$ and $n$. Given that $(1 - 2\rho)x + C(\rho, n)$ is nondecreasing with respect to $x \in \mathbb{R}$ when $0 \leq \rho < 0.5$ is held constant, $g_1$ remains the best feature under noise (i.e. has the highest unnormalized AUC value), and the feature ranking is maintained. Thus, the set of good features is the set of features $g_j$, where each $g_j$ satisfies the following:

$$\mathbb{E}_{S_\rho} \overline{AUC}_{S_\rho}(g_1) - E_{S_\rho}\overline{AUC}_{S_\rho}(g_j) \leq \gamma.$$

This is equivalent to

$$(1 - 2\rho)\overline{AUC}_S(g_1) + C(\rho, n) - (1 - 2\rho)\overline{AUC}_S(g_j) - C(\rho, n) \leq \gamma,$$

and

$$(1 - 2\rho)(\overline{AUC}_S(g_1) - \overline{AUC}_S(g_j)) \leq \gamma. \tag{11}$$

For ease of notation let, $a_j = \overline{AUC}_S(g_1) - \overline{AUC}_S(g_j)$. Let $k$ be the size of the set of good features under noise level $\rho_1$, so that $|G_{\mathbb{E}_{S_{\rho_1}} \overline{AUC}_{S_{\rho_1}}}(\mathcal{G}, \gamma)| = k$. Then, from (11) and the feature ranking, we have that:

$$(1 - 2\rho_1)a_1 \leq \cdots \leq (1 - 2\rho_1)a_k \leq \gamma < (1 - 2\rho_1)a_{k+1} \leq \cdots \leq (1 - 2\rho_1)a_p. \tag{12}$$

To show that $|G_{\mathbb{E}_{S_{\rho_2}} \overline{AUC}_{S_{\rho_2}}}(\mathcal{G}, \gamma)| \geq k + 1$, it suffices to show that $g_{k+1}$ is in the set of good features under noise level $\rho_2$, which is equivalent to the statement

$$(1 - 2\rho_2)a_{k+1} \leq \gamma.$$

Note that by the assumption of the corollary, for each $j = 1, \ldots, p - 1$, we have that

$$a_{j+1} - a_j = \overline{AUC}_S(g_j) - \overline{AUC}_S(g_{j+1}) \leq \delta.$$

Then, from (11) and (12) we get

$$(1 - 2\rho_1)a_{k+1} = (1 - 2\rho_1)a_k + (1 - 2\rho_1)(a_{k+1} - a_k) \leq \gamma + (1 - 2\rho_1)\delta,$$

which means that

$$a_{k+1} \leq \frac{\gamma}{1 - 2\rho_1} + \delta.$$

If $\rho_2 \geq \nu(\rho_1)$, then we have

$$
\begin{aligned}
(1 - 2\rho_2)a_{k+1} &\leq (1 - 2\rho_2)\left(\frac{\gamma}{1 - 2\rho_1} + \delta\right) \\
&\leq (1 - 2\nu(\rho_1))\left(\frac{\gamma}{1 - 2\rho_1} + \delta\right) \\
&\leq \left(1 - \left(1 - \frac{\gamma(1 - 2\rho_1)}{\gamma + \delta(1 - 2\rho_1)}\right)\right)\left(\frac{\gamma}{1 - 2\rho_1} + \delta\right) \\
&= \frac{\gamma(1 - 2\rho_1)}{\gamma + \delta(1 - 2\rho_1)} \cdot \left(\frac{\gamma}{1 - 2\rho_1} + \delta\right) \\
&= \frac{\gamma(1 - 2\rho_1)}{\gamma + \delta(1 - 2\rho_1)} \cdot \frac{\gamma + \delta(1 - 2\rho_1)}{1 - 2\rho_1} \\
&= \gamma.
\end{aligned}
\tag{13}
$$

Therefore, we proved that $(1 - 2\rho_2)a_{k+1} \leq \gamma$, which means that $|G_{\mathbb{E}_{S_{\rho_2}} \overline{AUC}_{S_{\rho_2}}}(\mathcal{G}, \gamma)| \geq k + 1$ and consequently $|G_{\mathbb{E}_{S_{\rho_1}} \overline{AUC}_{S_{\rho_1}}}(\mathcal{G}, \gamma)| < |G_{\mathbb{E}_{S_{\rho_2}} \overline{AUC}_{S_{\rho_2}}}(\mathcal{G}, \gamma)|$ as claimed. □

## F  Proof of Theorem 8

**Theorem 8** (Empirical Rashomon ratio increases when the set of good features increases). *For a dataset $S = X \times Y$ with binary feature matrix $X \in \{-1, 1\}^{n \times p}$, consider a hypothesis space $\mathcal{F}_d$ of fully grown trees of depth $d$. Consider uniform random label noise with noise parameter $\rho$. Let $\mathcal{H}_{set_{\mathbb{E}_{S_\rho}}}(\mathcal{F}_d)$ denote the set of grounded models, based on the set of good features $G_{\mathbb{E}_{S_\rho} \overline{AUC}_{S_\rho}}(\mathcal{G}, \gamma)$. Under the assumptions of Corollary 7 on the set of good features, the fraction of grounded models increases with uniform random label noise. More formally, for $\rho_2 \geq \nu(\rho_1)$,*

$$\left| \mathcal{H}_{set_{\mathbb{E}_{S_{\rho_1}}}}(\mathcal{F}_d) \right| < \left| \mathcal{H}_{set_{\mathbb{E}_{S_{\rho_2}}}}(\mathcal{F}_d) \right| \text{ and } \mathcal{H}_{ratio_{\mathbb{E}_{S_{\rho_1}}}}(\mathcal{F}_d) < \mathcal{H}_{ratio_{\mathbb{E}_{S_{\rho_2}}}}(\mathcal{F}_d).$$

*Proof.* As in Proposition 6 of Semenova et al. [2023], the hypothesis space of fully-grown trees of depth $d$ contains

$$|\mathcal{F}_d| = 2^{2^d} \prod_{k=1}^{d} (p - k + 1)^{2^{k-1}}$$

trees, where symmetric trees are not included, meaning that split = 0 is on the left and 1 is on the right.

The trees in the set of grounded models must have at least one good feature. Moreover, the labels in the set of grounded models are assigned to the trees based on the data, meaning that it contains

$$|\mathcal{H}_{set_{\mathbb{E}_{S_\rho}}}| = \prod_{k=1}^{d} (p - k + 1)^{2^{k-1}} - \prod_{k=1}^{d} (|p_{\mathbb{E}_{S_\rho}}^{bad}| - k + 1)^{2^{k-1}}$$

trees, where $|p_{\mathbb{E}_{S_\rho}}^{bad}| = p - |G_{\mathbb{E}_{S_\rho} \overline{AUC}_{S_\rho}}(\mathcal{G}, \gamma)|$. Therefore the fraction of grounded models is:

$$
\begin{aligned}
H_{ratio_{\mathbb{E}_{S_\rho}}}(\mathcal{F}_d^p) = \frac{|\mathcal{H}_{set_{\mathbb{E}_{S_\rho}}}(\mathcal{F}_d^p)|}{|\mathcal{F}_d^p|} &= \frac{\prod_{k=1}^{d} (p - k + 1)^{2^{k-1}} - \prod_{k=1}^{d} (|p_{\mathbb{E}_{S_\rho}}^{bad}| - k + 1)^{2^{k-1}}}{2^{2^d} \prod_{k=1}^{d} (p - k + 1)^{2^{k-1}}} \\
&= \frac{1}{2^{2^d}} \left( 1 - \frac{\prod_{k=1}^{d} (|p_{\mathbb{E}_{S_\rho}}^{bad}| - k + 1)^{2^{k-1}}}{\prod_{k=1}^{d} (p - k + 1)^{2^{k-1}}} \right) \\
&= \frac{1}{2^{2^d}} \left( 1 - \prod_{k=1}^{d} \left( \frac{|p_{\mathbb{E}_{S_\rho}}^{bad}| - k + 1}{p - k + 1} \right)^{2^{k-1}} \right).
\end{aligned}
$$

For noise levels $\rho_1, \rho_2$ from Corollary 7 if $\rho_2 \geq \nu(\rho_1)$, then $|G_{\mathbb{E}_{S_{\rho_1}} \overline{AUC}_{S_{\rho_1}}}(\mathcal{G}, \gamma)| < |G_{\mathbb{E}_{S_{\rho_2}} \overline{AUC}_{S_{\rho_2}}}(\mathcal{G}, \gamma)|$, and $|p_{\mathbb{E}_{S_{\rho_1}}}^{bad}| > |p_{\mathbb{E}_{S_{\rho_2}}}^{bad}|$. Therefore,

$$
\begin{aligned}
|\mathcal{H}_{set_{\mathbb{E}_{S_{\rho_1}}}}(\mathcal{F}_d^p)| &= \prod_{k=1}^{d} (p - k + 1)^{2^{k-1}} - \prod_{k=1}^{d} (|p_{\mathbb{E}_{S_{\rho_1}}}^{bad}| - k + 1)^{2^{k-1}} \\
&< \prod_{k=1}^{d} (p - k + 1)^{2^{k-1}} - \prod_{k=1}^{d} (|p_{\mathbb{E}_{S_{\rho_2}}}^{bad}| - k + 1)^{2^{k-1}} = |\mathcal{H}_{set_{\mathbb{E}_{S_{\rho_2}}}}(\mathcal{F}_d^p)|,
\end{aligned}
$$

and

$$H_{ratio_{\mathbb{E}_{S_{\rho_1}}}}(\mathcal{F}_d^p) = \frac{1}{2^{2^d}} \left( 1 - \prod_{k=1}^{d} \left( \frac{|p_{\mathbb{E}_{S_{\rho_1}}}^{bad}| - k + 1}{p - k + 1} \right)^{2^{k-1}} \right)$$

$$< \frac{1}{2^{2d}} \left( 1 - \prod_{k=1}^{d} \left( \frac{|p_{\mathbb{E}_{S_{\rho_2}}}^{bad}| - k + 1}{p - k + 1} \right)^{2^{k-1}} \right) = H_{ratio_{\mathbb{E}_{S_{\rho_2}}}}(\mathcal{F}_d^p).$$

Thus, with noise, the size of the set and fraction of grounded models increases. $\qquad\square$

# G  Proof of Theorem 9

**Theorem 9** (Exponential loss under additive attribute noise). *Consider the dataset $S$ and a hypothesis space $\mathcal{F}$ of linear models, $\mathcal{F} = \{f = \omega^T x, \omega \in \mathbb{R}^p\}$. For a given model $f \in \mathcal{F}$, consider the exponential loss $L_S(f) = \frac{1}{n} \sum_{i=1}^{n} e^{-y_i \omega^T x_i}$. Let $\epsilon_i$, such that $\epsilon_i \sim \mathcal{N}(\bar{0}, \sigma^2 I)$ ($\sigma > 0$, $I$ is identity matrix), be i.i.d. noise vectors added to every sample: $x_i' = x_i + \epsilon_i$. If $\mathbb{E}_\epsilon L_{S_{\epsilon(\sigma)}}(f)$ is the expected exponential loss under additive Gaussian noise, then*

$$\mathbb{E}_\epsilon L_{S_{\epsilon(\sigma)}}(f) = L_S(f) \cdot e^{\frac{\sigma^2}{2}\|\omega\|_2^2},$$

*where for simplicity we denote $\mathbb{E}_{\epsilon_1,\dots,\epsilon_n \sim \mathcal{N}(\bar{0}, \sigma^2 I)}$ as $\mathbb{E}_\epsilon$.*

*Proof.* We can write the following:

$$\mathbb{E}_\varepsilon L_{S_{\varepsilon(\sigma)}}(\omega) = \frac{1}{n} \sum_{i=1}^{n} \mathbb{E}_\varepsilon e^{-y_i \omega^T (x_i + \varepsilon)}$$

$$= \frac{1}{n} \sum_{i=1}^{n} e^{-y_i \omega^T x_i} \mathbb{E}_\varepsilon [e^{-y_i \omega^T \varepsilon}]$$

$$= \frac{1}{n} \sum_{i=1}^{n} e^{-y_i \omega^T x_i} \mathbb{E}_\varepsilon [e^{-\omega^T \varepsilon}]$$

$$= \mathbb{E}_\varepsilon [e^{-\omega^T \varepsilon}] \cdot \frac{1}{n} \sum_{i=1}^{n} e^{-y_i \omega^T x_i}$$

$$= L_S(\omega) \cdot \mathbb{E}_\varepsilon [e^{-\omega^T \varepsilon}]$$

From here, note that for any multivariate Gaussian random variable $\delta \sim \mathcal{N}(\mu, \Sigma)$ and matrix $A$, we have that $A\delta \sim \mathcal{N}(A\mu, A\Sigma A^T)$. Repeated usage of this identity yields

$$\mathbb{E}_\varepsilon [e^{-\omega^T \varepsilon}] = \mathbb{E}_{\delta \sim \mathcal{N}(\bar{0}, I)}[e^{-\sigma \omega^T \delta}] = \mathbb{E}_{\delta \sim \mathcal{N}(0, \|\omega\|_2^2)}[e^{-\sigma \delta}] = \mathbb{E}_{\delta \sim \mathcal{N}(0,1)}[e^{-\sigma \cdot \|\omega\|_2 \cdot \delta}]$$

We can directly compute the latter term by integrating. For any $k$,

$$\mathbb{E}_{\delta \sim \mathcal{N}(0,1)}[e^{k\delta}] = \int_{-\infty}^{\infty} \frac{1}{\sqrt{2\pi}} e^{-\frac{1}{2}x^2} \cdot e^{kx} dx$$

$$= \int_{-\infty}^{\infty} \frac{1}{\sqrt{2\pi}} e^{-\frac{1}{2}(x-k)^2 + \frac{1}{2}k^2} dx$$

$$= e^{\frac{1}{2}k^2} \int_{-\infty}^{\infty} \frac{1}{\sqrt{2\pi}} e^{-\frac{1}{2}(x-k)^2} dx$$

$$= e^{\frac{1}{2}k^2}$$

If we set $k = -\sigma \cdot \|\omega\|_2$, then we obtain the claim. $\qquad\square$

Note that there is an immediate generalization of Theorem 9 to when the standard deviation of noise applied is unequal across each feature. If feature $j$ receives independent Gaussian noise with standard deviation $\sigma_j$, then

$$\mathbb{E}_\varepsilon L_{S_{\varepsilon(\sigma)}}(\omega) = L_S(f) \cdot \exp\left( \frac{1}{2} \sum_{j=1}^{p} \sigma_j^2 \omega_j^2 \right)$$

We can observe that a higher degree of noise of a feature corresponds to a greater degree of regularization of the corresponding weight. In particular, if a given feature is not subject to additive Gaussian noise, then the weight corresponding to that feature is not regularized at all.

| Data Name | # samples | #features | Processing notes |
|---|---|---|---|
| Amsterdam (Recidivism) | 20000 | 9 | |
| Broward County (Recidivism) | 1954 | 16 | |
| NIJ Recidivism Challenge | 16264 | 15 | Fill missing prison offenses with 'Unknown' |
| Australian Credit | 640 | 15 | Outliers in columns 'A13' and 'A14' removed |
| German Credit | 1000 | 7 | |
| GiveMeSomeCredit | 18040 | 7 | Downsample to balance classes |
| Polish Companies | 1597 | 26 | Downsample to balance classes |
| Iranian Churn | 3150 | 8 | |
| Telco Churn | 7032 | 12 | |
| Occupancy Detection | 20560 | 26 | Drop 'date' column entirely |
| Bank-full | 9042 | 7 | Downsample to balance classes |
| Banknote Authentication | 1372 | 4 | |
| COMPAS (Recidivism) | 6907 | 13 | Same as [Xin et al., 2022, Semenova et al., 2023] |
| FICO (Credit) | 10459 | 18 | Same as [Xin et al., 2022, Semenova et al., 2023] |
| monks1 | 169 | 12 | Same as [Xin et al., 2022, Semenova et al., 2023] |
| monks2 | 124 | 12 | Same as [Xin et al., 2022, Semenova et al., 2023] |
| monks3 | 122 | 12 | Same as [Xin et al., 2022, Semenova et al., 2023] |
| Breast Cancer Wisconsin | 699 | 11 | Same as [Xin et al., 2022, Semenova et al., 2023] |
| Car Evaluation | 1728 | 16 | Same as [Xin et al., 2022, Semenova et al., 2023] |
| bar | 1913 | 16 | Same as [Xin et al., 2022, Semenova et al., 2023] |
| bar7 | 1913 | 15 | Same as [Xin et al., 2022, Semenova et al., 2023] |
| Carryout Takeaway | 2280 | 16 | Same as [Xin et al., 2022, Semenova et al., 2023] |
| Coffee House | 3816 | 16 | Same as [Xin et al., 2022, Semenova et al., 2023] |
| Restaurant 20 | 2653 | 16 | Same as [Xin et al., 2022, Semenova et al., 2023] |

Table 2: Summary count statistics of all datasets after preprocessing

## H  Datasets

In Table 2, we provide the description of the datasets used in this paper and pre-processing steps. All categorical variables were one-hot encoded, and all continuous data was binned into a one-hot binary encoding using thresholds determined by the GOSDT-guesses algorithm [McTavish et al., 2022]. All rows with missing values were removed from the datasets unless otherwise indicated in the processing notes. The GOSDT-guesses algorithm was also used to perform feature selection. The dataset statistics presented in Table 2 reflect the data used in the experiments after all pre-processing, including feature selection. When necessary, we downsampled to balance classes to avoid trivial optimal sparse decision trees.

## I  Additional Experiments

### I.1  The Complexity of Models in the Expected Empirical Rashomon Set

We now present an experiment demonstrating the results of Section 4.2, where we showed that the complexity of models in the true Rashomon set tends to decrease with the injection of label noise. In the experimental setting, we may only observe samples from the true data distribution, and thus we only have access to empirical Rashomon sets. To simulate the Rashomon set of models trained on the expected distribution of datasets under label noise, 150 noise draws were independently sampled then concatenated for every dataset to create a training set for sparse decision trees. We used a multiplicative Rashomon parameter of $0.05$ and a regularization parameter of $0.02$ throughout these experiments. Figure 3 shows how the discrete probability distribution of the number of leaves of models in the expected empirical Rashomon set skews simpler as noise is injected into the data. This result holds both for the aggregated number of leaves in all of our datasets (a), in recidivism and credit default datasets (b), and in individual datasets, some of which are displayed in Figure 5

### I.2  Model Simplification Under Expected Label Noise

**0-1 Loss and Sparse Decision Trees**  We now discuss the set-up of the experiment conducted to review Theorems 1 and 2. This experiment was conducted with sparse optimal decision trees trained on a subset of the complete feature set, with continuous features binarized via the binning procedure of the GOSDT-guesses algorithm [McTavish et al., 2022]. Throughout the experiment, we allowed a maximum depth of 5, which we did not find to be a limiting factor. Each dataset is split into a training set and a test set, where the training set consists of roughly 80% of the data. Before injecting

| Data Name | License | Citation |
|---|---|---|
| Amsterdam (Recidivism) | DANS | Tollenaar and Heijden [2013] |
| Broward County (Recidivism) | Publicly Available | Wang et al. [2023] |
| NIJ Recidivism Challenge | Publicly Available | NIJ [2011] |
| Australian Credit | CC BY 4.0 | Quinlan [2006] |
| German Credit | CC BY 4.0 | Hofmann [1994] |
| GiveMeSomeCredit | Kaggle Competition Rules | Credit Fusion [2011] |
| Polish Companies | CC BY 4.0 | Tomczak [2016] |
| Iranian Churn | CC BY 4.0 | mis [2020] |
| Telco Churn | Publicly Available | Hao [2024] |
| Occupancy Detection | CC BY 4.0 | Candanedo [2016] |
| Bank-full | CC BY 4.0 | Moros et al. [2012] |
| Banknote Authentication | CC BY 4.0 | Lohweg [2013] |
| COMPAS (Recidivism) | Publicly Available | Mattu et al. [2016] |
| FICO (Credit) | FICO | FICO [2018] |
| monks1 | CC BY 4.0 | Wnek [1992] |
| monks2 | CC BY 4.0 | Wnek [1992] |
| monks3 | CC BY 4.0 | Wnek [1992] |
| Breast Cancer Wisconsin | CC BY 4.0 | Wolberg [1992] |
| Car Evaluation | CC BY 4.0 | Bohanec [1997] |
| bar | CC BY 4.0 | Wang et al. [2020] |
| bar7 | CC BY 4.0 | Wang et al. [2020] |
| Carryout Takeaway | CC BY 4.0 | Wang et al. [2020] |
| Coffee House | CC BY 4.0 | Wang et al. [2020] |
| Restaurant 20 | CC BY 4.0 | Wang et al. [2020] |

Table 3: Licensing and Data Source Information for all Datasets

any noise, the optimal value $\lambda$ of the regularization parameter for the GOSDT algorithm was chosen using 5-fold cross validation on the training set. Then, for 51 parameters $\rho$ for label noise linearly spaced between 0 and 0.3, inclusive, we run the following procedure:

1. Sample 250 i.i.d. draws of random label noise with parameter $\rho$.

2. Stack all of these draws of random label noise into a single dataset.

3. Find the optimal sparse decision tree on this new, concatenated dataset.

4. Report the accuracy of the tree on its noisy training set (green), on the original (cleaner) training set (orange), and on the held out test set (blue).

5. Report the number of leaves (blue) of the optimal model trained on the noisy concatenated dataset.

6. Separately find the optimal sparse decision tree over the original training set with regularization parameter $\lambda/(1 - 2\rho)$, and report the number of leaves in this model (orange).

7. Compute the upper bound from Corollary 10 (green) based on the loss difference between the noisy optimal model and the baseline optimal model on the original data (difference in orange accuracy line from its initial value).

Complete results of these experiments for all datasets are shown in Figure 2, 6, and 7.

**Exponential Loss and Linear Models**  For the experiment corresponding to Theorem 9, we use a simpler setup to demonstrate that additive noise has a regularizing effect. We seek to optimize the exponential loss of a given binary classification dataset over the set of linear models. To do this, we directly use gradient descent with the exponential loss. We optimize for 1000 epochs with an initial learning rate of 0.1 with the ADAM optimizer [Kingma and Ba, 2015]. We furthermore decrease the learning rate by a factor of 0.3 if the exponential loss plateaus for over 50 epochs. Using this optimization procedure, our experiment is as follows:

1. Sample 100 i.i.d. draws of additive Gaussian noise with standard deviation $\sigma$ on all continuous features.

2. For each noisy draw, find the optimal linear model using the gradient descent procedure described above.

3. Report the sum of the squares of the weights of the optimal model corresponding to continuous noisy features.

## I.3 Empirical Evidence that the Set of Grounded Models Tend to Contain the Rashomon Set

Trees in the Rashomon set tend to rely on features with good signals; therefore, the set of grounded models tends to overlap significantly with the Rashomon set. We demonstrate this empirically based on 19 classification datasets in Table 4. To compute the set of good features, we used the AUC as a feature quality metric and set $\gamma$ to 0.05. To compute the Rashomon set, we considered the hypothesis space of sparse decision trees of depth 4 with a regularization parameter of 0.01. We used TreeFARMS and set the Rashomon parameter to 5%. For every tree in the Rashomon set, we verified whether it relies on at least one feature from the set of good features. The results are reported in the fourth column of Table 4. At least 95% of trees in every Rashomon set contains at least one feature from the set of good features.

| Data name | # of features | Size of the set of good features | Percentage of trees in the Rashomon set, that rely on at least one feature from the set of good features |
|---|---|---|---|
| Carryout Takeaway | 15 | 7 | 99.87% (18344/18368) |
| Amsterdam | 8 | 8 | 100% (1153/1153) |
| Coffee House | 15 | 8 | 100% (3483/3483) |
| Australian Credit | 14 | 1 | 100% (6296/6296) |
| COMPAS | 12 | 2 | 100% (19859/19859) |
| NIJ Recidivism | 14 | 4 | 99.98% (112065/112083) |
| Bank-full | 6 | 3 | 99.90% (1003/1004) |
| FICO | 17 | 2 | 100% (125809/125809) |
| Occupancy Detection | 25 | 13 | 99.99% (205762/205791) |
| bar7 | 14 | 2 | 100% (3939/3939) |
| German Credit | 6 | 2 | 95.21% (1807/1898) |
| Polish Companies | 25 | 10 | 99.55% (2107937/2117559) |
| bar | 15 | 6 | 100% (13950/13950) |
| GiveMeSomeCredit | 6 | 4 | 99.68% (1247/1251) |
| Restaurant 20 | 15 | 3 | 99.81% (112971/113189) |
| BCW-bin | 10 | 3 | 97.21 % (22552/23200) |
| Telco Churn | 11 | 5 | 95.82% (11830/12346) |
| Broward | 15 | 14 | 100% (416118/416119) |
| Iranian Churn | 7 | 4 | 95.16% (531/558) |

Table 4: The set of grounded models is likely to contain the Rashomon set for various datasets.

## I.4 Noise Distorts the Signal in Features and Increases the Size of the Set of Good Features

We now present an empirical demonstration of the effect of label noise on relative feature quality, which we measure as a feature's AUC with the labels (see Section 5.1). The uniformity assumption in our results in that section was designed to avoid situations where one feature is clearly more predictive than all of the others, which negates the effect of small amounts of label noise on the set of good features. Therefore, we designed our experiments to align with the assumptions in our theoretical results by intentionally removing features that were clear outliers in predictive quality. We are left with a somewhat uniform distribution of cleaner-data AUC values in Figure 4. With label noise, the distribution skews left as predicted – this indicates that the high-quality features degrade towards 0.5 AUC faster than low-quality features, as predicted in Theorem 5.

## I.5 Computation Resources

We performed experiments on Duke University's Computer Science Department cluster. We requested 200GB of shared total memory, and one compute core per dataset (23 datasets) so that we could run the experiments for each dataset in parallel. It took up to 10 hours to compute the optimal models for all different noise levels and draws (Figures 1a-b, ,2, 6, 7), and 2 hours to compute the simplicity of models in the Rashomon set (Figures 3, 5). For the experiments corresponding to Figure 1c, it took up to 3 hours per dataset to compute the optimal models for all different noise levels and draws.

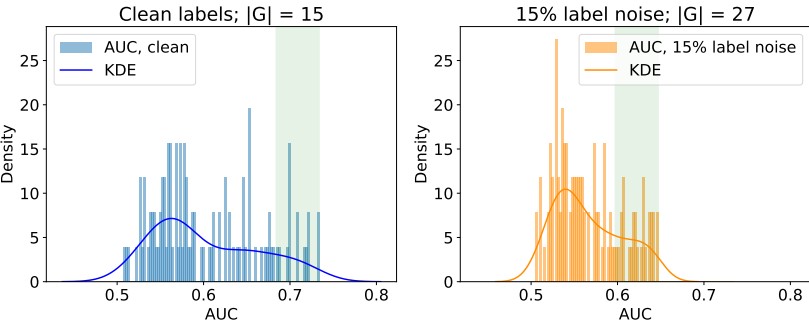

Figure 4: Feature AUC distribution before (left; blue) and after (right; orange) injecting 15% label noise, aggregated across 9 criminal recidivism and financial datasets with feature quality parameter $\gamma = 0.05$ (total of 109 features). All features degrade towards 0.5 AUC when label noise is introduced, and better features tend to degrade faster. This increases the size of the set of features within $\gamma$ AUC of the best feature (green shaded regions) from 15 features with original (cleaner) labels to 27 features with noisy labels, which aligns with our prediction in Corollary 7.

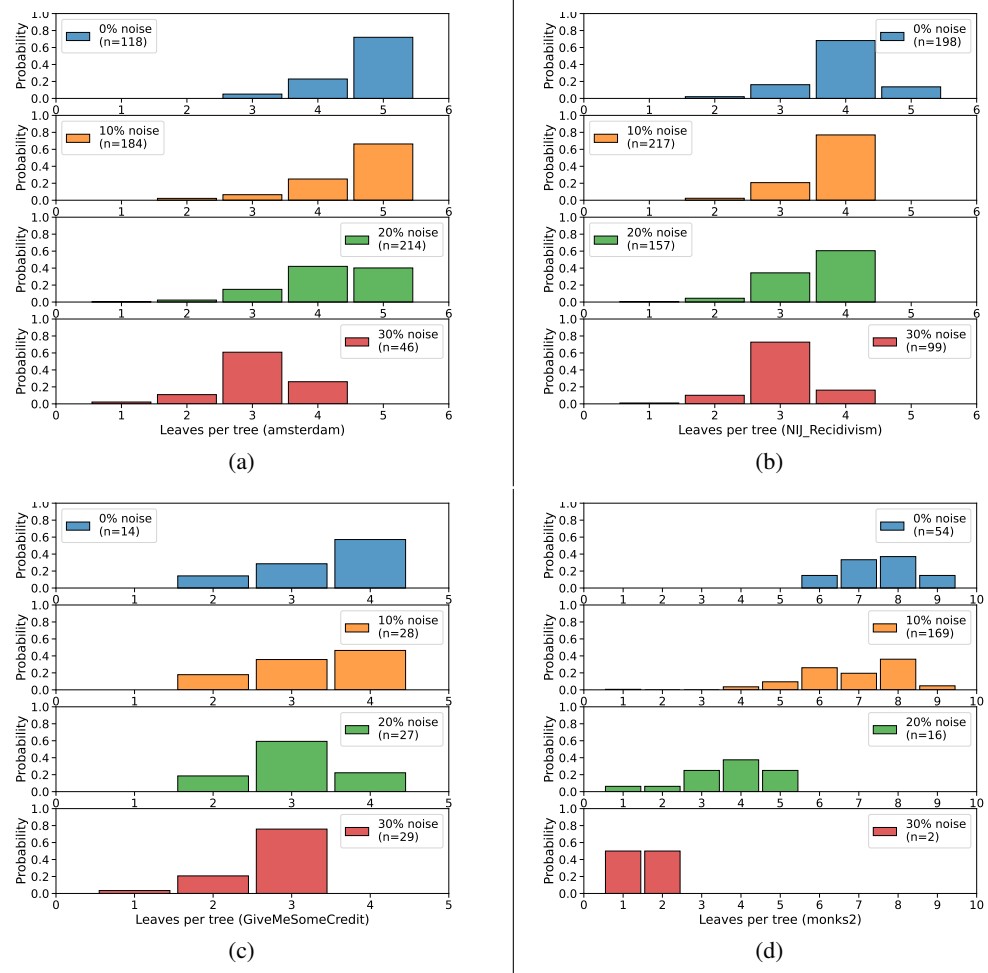

Figure 5: A selection of individual datasets from the findings in Figure 3. (a) is a recidivism dataset collected from the city of Amsterdam. (b) is a recidivism dataset from the NIJ's Recidivism Challenge. (c) is a credit default dataset from a Kaggle competition. (d) is a synthetic dataset. Datasets were chosen to represent an array of domains, including real-world and synthetic data.

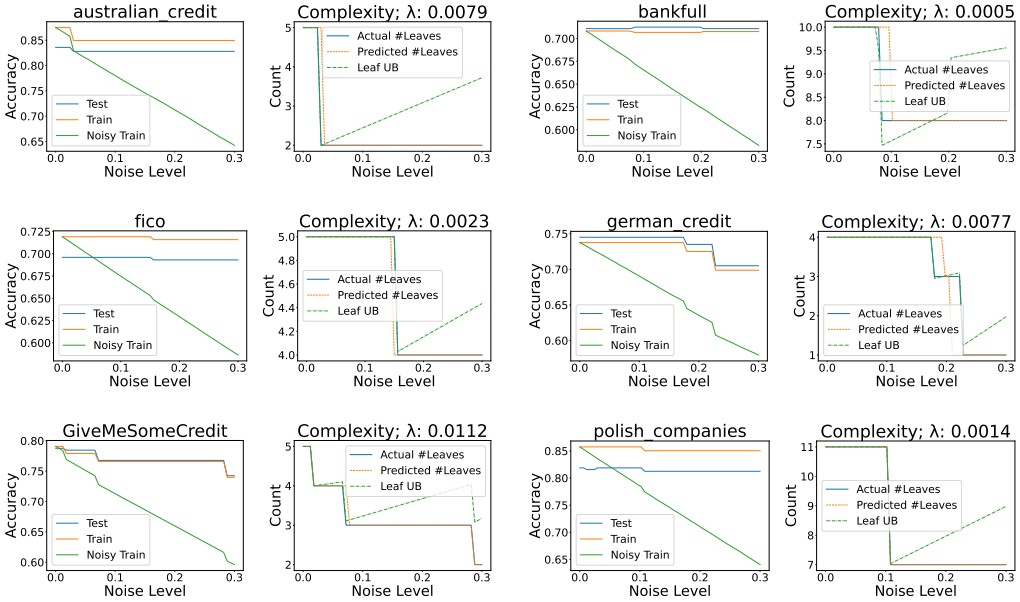

Figure 6: Experimental results for Section 4 on financial datasets. See experimental design in Appendix I.2 for a description of each line.

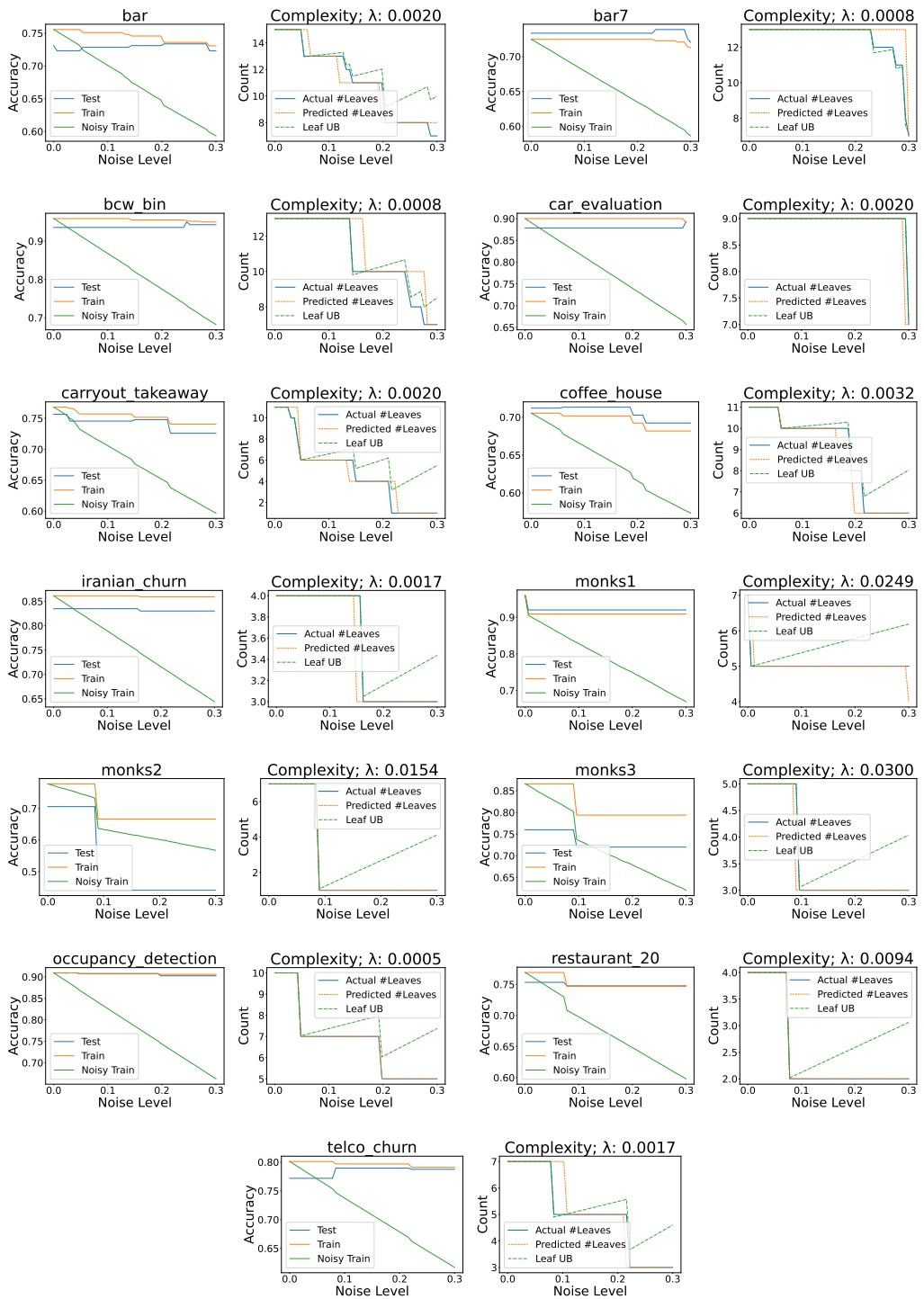

Figure 7: Experimental results for Section 4 on miscellaneous datasets. See experimental design in Appendix I.2 for a description of each line.

