# OpenReview forum: "Using Noise to Infer Aspects of Simplicity Without Learning"
_NeurIPS.cc/2024/Conference — NeurIPS 2024 poster_

### Official Review · Reviewer_5HeG · 2024-07-11

**Soundness:** 3
**Presentation:** 3
**Contribution:** 3
**Rating:** 6
**Confidence:** 3

**Summary:**

This paper explores the relationship between data noise and model simplicity across different hypothesis spaces, focusing on decision trees and linear models. The authors demonstrate that noise functions as an implicit regularizer for various noise models and show that Rashomon sets built from noisy data typically include simpler models than those from non-noisy data. Additionally, they reveal that noise broadens the set of "good" features and increases the number of models utilizing at least one good feature. This research offers theoretical assurances and practical insights for practitioners and policymakers on the likelihood of simple yet accurate machine learning models, based on understanding the noise levels in data generation processes.

**Strengths:**

- The paper delves deeply into the theoretical aspects of how noise influences model complexity, offering insights into the regularization effects of noise and its impact on the selection of simpler models in machine learning. This is interesting as it quantifies the effects of noise on model simplicity and expands the understanding of the Rashomon effect. Personally, I found the latter connection to be interesting in particular.

- The theoretical results are robust and well-presented. The proofs are detailed and appear sound.

**Weaknesses:**

- The connection between regularization and noise is well established using plenty of prior work (as demonstrated by the paper as part of the related work section). I feel like the paper sometimes stresses this insight too much as a novel idea. While the more detailed exploration of the connection to Rashomon sets is interesting and appears to be novel, I would strongly encourage the authors to not blur the lines between what has been shown in prior work already and what their contribution is. In my opinion, it should focus on the Rashomon set connection more strongly.

- The experimental section seems comparatively short and limited. Both the model types and datasets are very simple. For a more convincing evaluation, it would have been great to see a more comprehensive experimental panel. It seems to me that this claim of simpler models arising under noise would be especially interesting where the base models are already complex in nature (and not linear models or decision trees which can already be considered easily interpretable).

- The paper relies on limiting assumptions, such as uniform random label noise, which may not always hold in real-world scenarios. Discussing these limitations more explicitly would help in understanding the boundaries of the applicability of the results in more detail.

**Questions:**

- How robust are your theoretical results to the assumption of uniform random label noise?
- Do you have any preliminary insights (or hypotheses) about how your findings might generalize to other more complex models and datasets such as neural networks?

**Limitations:**

See weaknesses above.

---

> ### Author Rebuttal · Authors · 2024-08-07
>
> Thank you for the review. We address the questions point-by-point below.
>
> **W1. Connection to the previous works.**
> Theorems 1 and 2 do show that noise acts as an implicit regularizer (similar to Bishop, Training with noise is equivalent to Tikhonov regularization, 1995; Dhifallah and Lu, On the inherent regularization effects of noise injection during training, 2021). However, Theorem 1 quantifies the exact relationship between noise and regularization $(\lambda / (1-2p))$ for 0-1 loss. These results are *general to the regularization function and algorithm independent*. In comparison, there are a lot of works that depend on *using specific algorithms like SGD* and regularization functions such as the $\ell_2$ norm. That is, our results depend on the noise in the data, not the noise imposed onto the algorithm, like the recent literature on neural network convergence. Also, our results apply to discrete hypothesis spaces like decision trees and rule lists which have not, to our knowledge, been addressed by prior works.
>
>
> We will also cite more works on noise acting as an implicit regularizer in the related works and comment on our differences in the paper revision.
>
>
> **W2. Empirical results.**
> The phenomenon described in the paper is exactly the reason why we do not need complex models on tabular datasets. The datasets we used represent a wide array of tabular data problems, including (but not limited to) criminal justice and lending decisions, boolean circuits, and survey results on restaurant attendance. These datasets were not picked for convenience. Our results aim to help explain why we are able to find well-performing simple models on most noisy tabular decision problems. By simple model we mean a very sparse model, like a decision tree with five to seven leaves, or a scoring system with 10-20
> components, while compared to the more complex models such as the decision tree of depth ten ($\sim 1024$ leaves for binary, fully grown tree) or linear model with 100 components. In practice, such model size differences can be crucial for interpretability.
>
> As an initial experiment to show simplification of more complex models, we evaluated a CNN trained on CIFAR-10 with clean vs. noisy labels. We found that training on noisy labels resulted in significantly more near-zero weights (see global response file for details).
>
>
> **W3. Noise model.**
> In reality, many datasets contain a high level of label noise. Please note that our results do not preclude the original distribution from having some base (potentially non-uniform) noise. Our results apply when uniform label noise is added on top of some existing noise.
>
> To show that our results hold for other losses and noise models, we also considered Gaussian feature noise for classification with exponential loss in Section 6, and we generalized this result to non-uniform Gaussian feature noise in Appendix G. Overall, we expect that the paper's results will hold for other label noise models. For example, please see Figure 1 in the global rebuttal response, where we considered a new noise model with different noise rates for different groups of the population.
> As in the experiments in our paper, we found simpler optimal models under a form of non-uniform label noise.
>
> We will emphasize more in the paper the potential of future work exploring other, more complicated noise models.
>
>
>
> **Q1. Robustness to noise models.**
> Our proof techniques for Theorems 1 and 2 are not easily generalizable to other noise models. This is because in the proof we rely on the equivalence of optimization problems, which holds under uniform label noise, but does not happen, for example, when noise can change marginal label distributions differently.
> However, we expect the empirical behavior of decision tree optimization under non-uniform random label noise to be similar to uniform random label noise. This is because, in regularized hypothesis spaces, noise (uniform or not) cannot add any useful signal to data that would require a complex model; it can only destroy signal and make it more difficult to distinguish the true decision boundary. Please see Figure 1 in the global rebuttal response for more empirical evidence.
>
> **Q2. Generalization to neural networks.**
> Our results hold for any hypothesis space optimized over regularized 0-1 loss. We expect our findings to generalize somewhat to more complex models like neural networks optimized on other loss functions (like mean squared error or cross-entropy) based on initial empirical evidence (please see global response file). For some non-rigorous intuition for neural networks: if you represent a neural network as an embedding layer followed by a linear classifier, then our results apply directly to label noise in the final embedding before classifications (which may be carried through from noise in the inputs in some non-uniform way).
> In general, though, our work aims to discuss the likely existence of a simple and accurate interpretable model for tabular datasets. If such a model exists, this knowledge can help a practitioner (or policymaker) decide to search for a simple model instead of a complex neural network or tree ensemble, for example. For this reason, much of our work is focused on the simpler hypothesis spaces instead of on the more complex ones.
>
> Thank you again for the review, we'd be happy to engage in the discussion regarding any points mentioned.

---

> > ### Author Response · Authors · 2024-08-12
> >
> > Thank you again for your time and feedback on the paper. We are writing as the author-reviewer discussion period is ending soon. If you have any questions or comments, please let us know, as we would be happy to discuss them before the deadline.

---

> > > ### Comment · Reviewer_5HeG · 2024-08-12
> > > **Rebuttal response**
> > >
> > > I thank the authors for their response. I think my questions have been largely addressed. I have updated my score to reflect that.

---

### Official Review · Reviewer_R9J1 · 2024-07-12

**Soundness:** 4
**Presentation:** 4
**Contribution:** 4
**Rating:** 6
**Confidence:** 2

**Summary:**

This work sets out to the study the effect learning with noise entails from the perspective of model complexity. This paper invokes formalism from Rashomon sets and derives theoretical results that show an equivalence between noise (in the form of labels flipped) and regularization and in particular show that models that are trained on noisy data are necessarily underperforming on original losses. The theory is then applied to decision trees and linear models.

**Strengths:**

[+] A very important and open-ended problem is pursued, involving the essence behind generalization trade-offs in machine learning.

[+] The theoretical results are very general with no assumptions made on the hypothesis class (Theorem 1)

[+] The paper is well-written and very easily accessible to a large variety of audiences.

**Weaknesses:**

[-] Analysis is a bit limited to models where the trade-offs are already well-understood such as decision trees.

[-] The theory seems to be focused on regularization where a penalty is involved only.

**Questions:**

(1) Is there any intuition how this can be applied to mainstream regularization techniques like drop-out?
(2) While the theory is developed for general regularization penalties, can something more specific be said for specific choices?
(3) Can the Theorems be adopted to regularization schemes beyond those that involve a loss such as augmentation?

**Limitations:**

Yes

---

> ### Author Rebuttal · Authors · 2024-08-07
>
> Thank you for the review. We address the questions point-by-point below.
>
> **W1. Limitations of analysis**.
> We've been studying decision trees for over a decade and we are not convinced that the trade-off is, thus far, well understood. From our experience, often for tabular datasets, there exists a sparse model that performs as well as black boxes. We believe that this phenomenon requires further investigation into how sparse models we can actually find before spending a lot of engineering time and effort. Our work provides an answer to this question when data come from noisy data generation processes.
>
>
> **W2. The regularization function generalization**.
> Yes, in this paper we consider explicit regularization in the classic sense as a function
> added to the loss (see Tikhonov et.al., Solutions of ill-posed problems, 1977). Note that this technique encompasses an enormous number of methods.
> We would need to know what explicit or implicit regularization method you have in mind to offer more detailed comments.
>
> **Q1. Can we apply results to drop-outs?**  Dropout is a modification to neural network training which has a regularizing effect, but is not an explicit regularizer. Our results apply to explicit regularization penalties added to the objective function.
>
> Please note that in this paper, we mainly work with tabular datasets, where one does not usually benefit from using neural networks, so techniques like dropout do not apply. Tabular data are very common in real-world applications, especially in lending and criminal justice which we mainly considered in the paper.
>
>
> Note that for $\ell_2$-regularized neural nets on image data, in an initial experiment, we observed a significant regularizing effect of label noise. Please see the global rebuttal response for details of this experiment.
>
> **Q2. More specific results for the regularization**.
> For Theorems 1 and 2 we can use our results to understand specific hypothesis spaces, like decision trees with regularization on the number of leaves. The results in these theorems are tight with respect to the regularization function. For specific regularization functions and specific hypothesis spaces, it might be possible to get more measurable results on simplification (for example questions like how many leaves will the optimal tree have with 25\% intrinsic noise). We are indeed working towards these kinds of results.
>
>
> **Q3. Can we generalize to augmentation?**
> In the tabular domain with meaningful features, we do not know of many augmentation techniques except adding noise. From Sections 3 and 6, we know that our results will hold under such augmentation.
>
> Thank you again for the review, we'd be happy to engage in the discussion regarding any points mentioned.

---

> > ### Author Response · Authors · 2024-08-12
> >
> > Thank you once again for your feedback and time. As we approach the end of the author-reviewer discussion period, we wanted to check if our response has fully addressed your concerns. If you have any further comments or questions, please let us know. We would be happy to discuss them before the deadline.

---

### Official Review · Reviewer_HAB8 · 2024-07-13

**Soundness:** 4
**Presentation:** 3
**Contribution:** 3
**Rating:** 8
**Confidence:** 3

**Summary:**

The paper explores the role of dataset noise in the possibility of training simpler models. They show that more noisy settings are more likely to contain simpler models due to an increase in the regularization factor for learning algorithms that employ regularization. In the same setting, they also show that the optimal model under noise is simpler (or at least, not more complex) than the clean data and that the Rashomon is likely to contain simpler models when there is noise in the data. The authors complete their discussion by showing similar trends even in the absence of regularization, by viewing it from the perspective of ‘good features’. All theoretical claims are supported by empirical evidence.

**Strengths:**

- The paper is well presented and most theoretical discussions were easy to follow, given of course the page limit of a conference paper. For a paper as theoretical as this, I'd recommend proof sketches in the main paper to allow a better reading experience. However, the authors do a good job of wading through the claims even without them.
- The insights of the paper are highly motivating for the real-world applications of ML. Real-world data can be noisy, partly because of the noise in the data collection pipeline, but also partly because it is not possible to always predict certain events. As shown by the authors, the Rashomon set in such a world is more likely to contain simpler models, which is quite promising and creates better incentives to search for those simpler models.
- Most theoretical claims in the paper are supported by experiments towards the end of the paper. While I did go through the theoretical claims to the best of my expertise, having the empirical evidence to support that the expected behaviour is indeed present in practice is always good.

**Weaknesses:**

- At certain points in the paper, the authors claim more than they actually show through their proofs. For example, Theorem 3 says that models that enter the Rashomon set under noise (F_in) are likely to be simpler than the optimal model. The authors use this theorem to claim that the Rashomon set under noise would tend to contain less complex models. However, just theorem 3 is not enough to make such a claim. What about the models that leave the Rashomon set, i.e., F_out? Were they complex models, which would support the authors' claims? Or were they simple models, in which case maybe there is no straightforward expectation of whether the overall set has become simpler or complex? Another example is Theorem 1, which the authors prove for the distribution-based Rashomon sets, and then claim that the same can be extended to empirical Rashmon sets. However, their proof relies on a relationship between loss under noisy data vs clean data that they borrow from Semenove et al. This relationship does not directly transfer to the empirical setting, where an additional expectation term across different datasets sampled from the distribution is introduced. While these small issues do not directly impact the overall message of the paper, I would highly recommend that the authors be extra careful with the language they use and the claims they make. Edit after rebuttal: Acknowledged.
- The empirical results provided by the authors are very targeted towards only the final claims in their theoretical discussion. However, a more complete picture through the empirical results would have made the paper so much stronger. For instance, what are the accuracy scores for models with and without noise? How much does the accuracy suffer due to noise, and while the Rashomon sets do contain simpler models, do they still even have some predictive power left, or is it too noisy? What fractions of models belong to F_in, F_both, and F_out in practice? There are many other questions that I would have liked to have seen empirically. Although I do understand the choice to focus more on the theory, and this isn't a big weakness of the paper. Edit after rebuttal: The empirical results were present in the Appendix.

**Questions:**

Please see the weaknesses.

**Limitations:**

Please see the weaknesses.

---

> ### Author Rebuttal · Authors · 2024-08-07
>
> Thank you for the review. We appreciate your points and feedback.
>
> **W1. Theorem 3 conclusion**.
>
> Thank you for pointing this out.
>
> We expect $F_{out}$ to mostly contain more complex models, since more complex models are a "closer" fit (overfit) to the data and therefore perform worse when noise is injected. Our experimental results support this hypothesis for the distribution of model complexities (see Figure 2 and Figure 4 in the Appendix). We can also show theoretically that models in $F_{out}$ are more complex than the optimal model for the noisier data under similar assumptions as Theorem 3 (please see the global rebuttal response). We will include more explanation in the text around Theorem 3 about the complexity of models in $F_{out}$.
>
>
>
>
> For Theorem 1, we will add the language to address the limitation of the empirical setting.
>
> **W2. Empirical results.**
>
> Thank you for asking all these interesting questions! Please see answers to them below, which we believe can be inferred from Figures 4-7 in the Appendix. We will add more discussion of these points to the paper.
>
> "How much does the accuracy suffer due to noise?" -  The test accuracy usually remains relatively stable until a very destructive amount of noise (around 25\% in most cases) is injected into the training set (please see Figures 5-7 in the appendix).
>
> "While the Rashomon sets do contain simpler models,
> do they still even have some predictive power left, or is it too noisy?" - The test accuracy remains stable with a lower amount of noise (see Figures 5-7). Since Theorems 1 and 2 show equivalence of the noisy optimization problem to the clean problem with higher regularization, we expect that the optimal model should still have predictive power unless the noise is large enough to make the effective regularization dominate the misclassification error. The models in the Rashomon set are close in accuracy to the optimal model, so we expect them to also have predictive power.
>
> "What fractions of models belong to $F_{in}$,
> $F_{both}$, and $F_{out}$ in practice?" - One can infer an estimate of sizes of $F_{in}$,
> $F_{both}$, and $F_{out}$ from Figure 4 in the appendix by observing the shift in the distribution of model complexities in the Rashomon set to the left. Increases in bars correspond to $F_{in}$, decreases to $F_{out}$, and the rest to $F_{both}$. Behavior varies by dataset and noise level. To provide more specific numbers, for the COMPAS dataset with $\rho = 0.2$,  $F_{in}$ contains approximately 502 models (average number of leaves over $F_{in}$ is 4.98), $F_{out}$ has 593 models (average number of leaves over $F_{out}$ is 6.46), and $F_{both}$ has 2230 models (average number of leaves over $F_{both}$ is 5.41) (The Rashomon parameter is set to 0.03, decision tree depth to 4 and sparsity regularization to 0.01).
>
>
> Much of our empirical exploration is in the Appendix, due to space constraints. We will gladly expand on some empirical questions in the paper with an additional page if accepted.

---

> > ### Comment · Reviewer_HAB8 · 2024-08-09
> >
> > Thank you for the additional proofs and for highlighting the empirical trends I missed in the Appendix.
> >
> > The rebuttal is acknowledged, and I will raise my scores further to reflect it.

---

### Official Review · Reviewer_aFkG · 2024-07-17

**Soundness:** 3
**Presentation:** 2
**Contribution:** 2
**Rating:** 6
**Confidence:** 4

**Summary:**

This work uses Rashomen sets ($R_{set_D} (\mathcal{F}, \theta) = \{ f \in \mathcal{F} : Obj_D(f) \leq Obj_D(f^*_D) + \theta\}$) to understand how the complexity of the optimal classifier simplifies with random label noise and additive gaussian noise. They show that label noise has explicit and implicit regulatory effects. They contribute the following conclusions
1. (Explicit) The optimal classifier for the Error (0-1 loss) + Reg with random label noise is equal to solving the same problem without label noise but scaling up the regularization term by 1/(1-2p) where p is the probability of flipping the label.
     - As a result, the optimal classifier with random label noise achieves has higher Error and smaller Reg than without label noise.
     - Any model in the Rashomen set under label noise and not in the Rashomen set without label noise has smaller Reg than optimal model without label noise.
2. (Implicit) In the setting without Reg in Decision Trees. The "predictability" of a feature, measured by AUC, degrades with label noise at different rates. Good features with higher AUC lose signal faster than features with lower AUC. As a result, the set of "best" features increases and accordingly the set of "good" models also increases.
3. (Implicit) In the setting of exponential loss in linear models with additive gaussian noise, there is an implicit L2 weight regularization.

**Strengths:**

The paper is clearly written and technically solid. Experiments conducted in linear models and decision trees substantiate that label noise does indeed regularize models consistent with theoretical claims.

**Weaknesses:**

- My main concern is the related works section, which I think is missing a large portion of work that has studied this problem. The current paragraph called "Noise and Regularization" section could delve more into previous theoretical works on the implicit regularization of label noise and gaussian noise, and discuss less about designing robust loss functions if space does not allow, since designing label-noise methods is less relevant to the paper. For example,\
[1] Loucas Pillaud-Vivien, Julien Reygner, Nicolas Flammarion. Label noise (stochastic) gradient descent implicitly solves the Lasso for quadratic parametrization. 2022. \
[2] Jeff Z. HaoChen, Colin Wei, Jason D. Lee, Tengyu Ma. Shape Matters: Understanding the Implicit Bias of the Noise Covariance. 2020. \
[3] Alex Damian, Tengyu Ma, Jason D. Lee. Label Noise SGD Provably Prefers Flat Global Minimizers. 2021. \

- It also seems that the work largely extends upon a previous work Semenova et al. It may also be good to explicitly discuss what the additional contributions of the paper's theorems are on top of previous conclusions. such as Theorem 8 in Semenova.

- In Theorem 1 and 2, should there be some additional constraints on the regularization function? Such as nonnegative.

**Questions:**

See above

---

> ### Author Rebuttal · Authors · 2024-08-07
>
> Thank you for the review. We address the weaknesses point-by-point below.
>
> **W1: Related literature.** Thank you, we will definitely add more relevant literature on implicit regularization as we will gain an extra page of space if the paper gets accepted. We will cite suggested papers on noise's effect on stochastic gradient descent as well as others related to them. *While this prior work considers a specific algorithm (SGD), our analysis in Sections 3 and 6  is algorithm independent.* It instead connects noise and regularization. Results in Section 3 are general for any algorithm that optimizes 0-1 loss alongside any regularization function. And results in Section 6 hold for any algorithm that optimizes exponential loss with $\ell_2$ regularization.
>
> **W2: Connection to work of Semenova from FaccT.** For *non-regularized* hypothesis spaces: Theorem 8 in the paper of Semenova et.al., 2022 says that the Rashomon set does not decrease in size with more noise. In our paper, we provide stronger results. We illustrate theoretically that for the non-regularized case, both the Rashomon set and the Rashomon ratio tend to increase with noise due to the increase of the size of the set of good features. This result is also stronger than the result in the paper of Semenova et.al., 2023 which observed an increase in sets and ratios only empirically.
>
> For *regularized* hypothesis spaces: For the regularized case, we are different from both papers as we show how much regularization changes with noise. Additionally, in Theorem 3 we prove that the complexity of the models in the Rashomon set tends to decrease.
>
> We communicated these differences in the Introduction and we will emphasize them more in the theorems framing text during revision.
>
>
> **W3: Non-negativity of the regularization function.**
> Thank you for pointing this out. Indeed, in all our examples (such as decision trees and linear models) we considered non-negative regularization. However, Theorems 1 and 2 do not require a negative regularization function. We will point this out in the revised paper.
>
> Thank you again for the review, we'd be happy to engage in the discussion regarding any points mentioned.

---

> > ### Comment · Reviewer_aFkG · 2024-08-10
> > **Thanks**
> >
> > Thank you for the clarification! My questions have been answered, and I've raised my score.

---

### Author Rebuttal · Authors · 2024-08-07

We thank all the reviewers for the reviews. Below, we provide proof that models that exit the clean Rashomon set are complex and an initial experiment on neural networks under random label noise. In the response file, we also include empirical analysis for the mixed label noise model.

**Models in $F_{out}$ in Section 4 are complex.** In the setting of Section 4, we empirically verified that models that exit the Rashomon set are complex models that potentially will overfit the noisier data in Figure 4. Here, we show mathematically that models in $F_{out}$ tend to be more complex (have a higher regularization penalty) than the empirical risk minimizer over the noisier data in Theorem 1.

**Theorem** (Models that exit the clean true Rashomon set are complex). Consider true data distribution $D$, 0-1 loss function, regularization $R(\cdot)$ and regularization parameter $\lambda$. Consider also uniform label noise, where each label is flipped independently with probability
$\rho < \frac{1}{2}$. Let $D_{\rho}$ be the noisier data distribution. If the optimal model over noisy data distribution $D$ is not in the clean true Rashomon set with Rashomon parameter $2\rho\theta$, i.e., $f_{D_{\rho}}^* \not\in R_{set_{D}}(\mathcal{F},2\rho\theta)\subset R_{set_{D}}(\mathcal{F},\theta)$ (note that this is a symmetric assumption to the assumption in the Theorem 3 in the paper), then every model from $F_{out}$ that exits the clean true Rashomon set $R_{set_{D}}(\mathcal{F},\theta)$ is more complex than $f_{D_{\rho}}^*$:
$$\forall \tilde{f} \in F_{out}: R(\tilde{f}) > R(f_{D_{\rho}}^*) + 2(1-\rho)\frac{\theta}{\lambda}.$$

**Neural networks experiments.** To investigate the effect of random label noise, in the distributional sense explored in the paper, we trained a CNN, utilizing cross-entropy loss and $\ell_2$ regularization, on the CIFAR-10 dataset with and without noise. The noisy data was constructed by randomly sampling label noise 100 times and stacking these samples, as in the experiments in the paper. We only added noise to the training data - the test data is clean. The architecture had a total of 62006 trainable parameters - to measure simplicity, we counted the number of parameters less than 1e-5 (near-zero parameters). For the model trained on the clean data, we found 37958 near-zero parameters. The model trained on the stacked noisy data, in contrast, had 51614 near-zero parameters. Both networks had around 60\% test accuracy. This, empirically, is an example of the regularizing effect of label noise, even for neural networks trained on cross-entropy loss. We present this experiment as initial evidence that our results can apply to more complex hypothesis spaces and loss functions.

---

### Decision · Program_Chairs · 2024-09-25

**Decision:**

Accept (poster)

**Comment:**

This paper studies the interaction between data set noise and model complexity, specifically in the context of decision trees and linear models. First, the papers proves for some particular settings that noise acts as an implicit regularizer and leads to simpler models. In addition, the paper shows that under noise, the set of near-optimal models consist of simpler models than in the non-noisy setting. In the unregularized setting, the paper studies the number of "good" features changes with noise. The results are then validated empirically.

Reviewers unanimously recommended this paper for acceptance. Reviewers have suggested that the paper should expand its discussion of related work (and make it more explicitly clear how this contribution differs from existing work), clarify and discuss the assumptions needed for the theorems, and incorporate more of the empirical results in the appendix into the main paper. We strongly encourage the authors to incorporate these revisions into the final version of the manuscript.